# LAST LAYER RE-TRAINING IS SUFFICIENT FOR ROBUSTNESS TO SPURIOUS CORRELATIONS

**Polina Kirichenko**[*]
New York University

**Pavel Izmailov**[*]
New York University

**Andrew Gordon Wilson**
New York University

## ABSTRACT

Neural network classifiers can largely rely on simple spurious features, such as backgrounds, to make predictions. However, even in these cases, we show that they still often learn core features associated with the desired attributes of the data, contrary to recent findings. Inspired by this insight, we demonstrate that simple last layer retraining can match or outperform state-of-the-art approaches on spurious correlation benchmarks, but with profoundly lower complexity and computational expenses. Moreover, we show that last layer retraining on large ImageNet-trained models can also significantly reduce reliance on background and texture information, improving robustness to covariate shift, after only minutes of training on a single GPU.

## 1 INTRODUCTION

Realistic datasets in deep learning are riddled with *spurious correlations* — patterns that are predictive of the target in the train data, but that are irrelevant to the true labeling function. For example, most of the images labeled as butterfly on ImageNet also show flowers (Singla & Feizi, 2021), and most of the images labeled as tench show a fisherman holding the tench (Brendel & Bethge, 2019). Deep neural networks rely on these spurious features, and consequently degrade in performance when tested on datapoints where the spurious correlations break, for example, on images with unusual background contexts (Geirhos et al., 2020; Rosenfeld et al., 2018; Beery et al., 2018). In an especially alarming example, CNNs trained to recognize pneumonia were shown to rely on hospital-specific metal tokens in the chest X-ray scans, instead of features relevant to pneumonia (Zech et al., 2018).

In this paper, we investigate what features are in fact learned on datasets with spurious correlations. We find that even when neural networks appear to heavily rely on spurious features and perform poorly on minority groups where the spurious correlation is broken, they still learn the core features sufficiently well. These core features, associated with the semantic structure of the problem, are learned even in cases when the spurious features are much simpler than the core features (see Section 4.2) and in some cases even when no minority group examples are present in the training data! While both the relevant and spurious features are learned, the spurious features can be highly weighted in the final classification layer of the model, leading to poor predictions on the minority groups.

Inspired by these observations, we propose *Deep Feature Reweighting* (DFR), a simple and effective method for improving worst-group accuracy of neural networks in the presence of spurious features. We illustrate DFR in Figure 1. In DFR, we simply retrain the last layer of a classification model trained with standard Empirical Risk Minimization (ERM), using a small set of *reweighting* data where the spurious correlation does not hold. DFR achieves state-of-the-art performance on popular spurious correlation benchmarks by simply reweighting the features of a trained ERM classifier, with no need to re-train the feature extractor. Moreover, we show that DFR can be used to reduce reliance on background and texture information and improve robustness to certain types of covariate shift in large-scale models trained on ImageNet, by simply retraining the last layer of these models. We note that the reason DFR can be so successful is because standard neural networks *are* in fact learning core features, even if they do not primarily rely on these features to make predictions, contrary to recent findings (Hermann & Lampinen, 2020; Shah et al., 2020). Since DFR only requires retraining a last layer, amounting to logistic regression, it is extremely simple, easy to tune and computationally inexpensive relative to the alternatives, yet can provide state-of-the-art performance. Indeed, DFR can reduce texture bias and improve robustness of large ImageNet trained models, in only minutes on a single GPU. Our code is available at `github.com/PolinaKirichenko/deep_feature_reweighting`.

---

[*]Equal contribution.

Figure 1: **Deep feature reweighting (DFR).** An illustration of the DFR method on the Waterbirds dataset, where the background (BG) is spuriously correlated with the foreground (FG). Standard ERM classifiers learn both features relevant to the background and the foreground, and weight them in a way that the model performs poorly on images with confusing backgrounds. With DFR, we simply reweight these features by retraining the last linear layer on a small dataset where the backgrounds are not spuriously correlated with the foreground. The resulting DFR model primarily relies on the foreground, and performs much better on images with confusing backgrounds.

## 2 PROBLEM SETTING

We consider classification problems, where we assume that the data consists of several *groups* $\mathcal{G}_i$, which are often defined by a combination of a label and spurious attribute. Each group has its own data distribution $p_i(x, y)$, and the training data distribution is a mixture of the group distributions $p(x, y) = \sum_i \alpha_i p_i(x, y)$, where $\alpha_i$ is the proportion of group $\mathcal{G}_i$ in the data. For example, in the Waterbirds dataset (Sagawa et al., 2019), the task is to classify whether an image shows a landbird or a waterbird. The groups correspond to images of waterbirds on water background ($\mathcal{G}_1$), waterbirds on land background ($\mathcal{G}_2$), landbirds on water background ($\mathcal{G}_3$) and landbirds on land background ($\mathcal{G}_4$). See Figure 6 for a visual description of the Waterbirds data. We will consider the scenario when the groups are not equally represented in the data: for example, on Waterbirds the sizes of the groups are 3498, 184, 56 and 1057, respectively. The larger groups $\mathcal{G}_1, \mathcal{G}_4$ are referred to as *majority groups* and the smaller $\mathcal{G}_2, \mathcal{G}_3$ are referred to as minority groups. As a result of this heavy imbalance, the background becomes a *spurious feature*, i.e. it is a feature that is correlated with the target on the train data, but it is not predictive of the target on the minority groups. Throughout the paper we will discuss multiple examples of spurious correlations in both natural and synthetic datasets. In this paper, we study the effect of spurious correlations on the features learned by standard neural networks, and based on our findings propose a simple way of reducing the reliance on spurious features assuming access to a small set of data where the groups are equally represented.

## 3 RELATED WORK

**Feature learning in the presence of spurious correlations.** The poor performance of neural networks on datasets with spurious correlations inspired research in understanding when and how the spurious features are learned. Geirhos et al. (2020) provide a detailed survey of the results in this area. Several works explore the behavior of maximum-margin classifiers, SGD training dynamics and inductive biases of neural network models in the presence of spurious features (Nagarajan et al., 2020; Pezeshki et al., 2021; Rahaman et al., 2019). Shah et al. (2020) show empirically that in certain scenarios neural networks can suffer from *extreme simplicity bias* and rely on simple spurious features, while ignoring the core features; in Section 4.2 we revisit these problems and provide further discussion. Hermann & Lampinen (2020) and Jacobsen et al. (2018) also show synthetic and natural examples, where neural networks ignore relevant features, and Scimeca et al. (2021) explore which types of shortcuts are more likely to be learned. Kolesnikov & Lampert (2016) on the other hand show that on realistic datasets core and spurious features can often be distinguished from the latent representations learned by a neural network in the context of object localization.

**Group robustness.** The methods achieving the best worst-group performance typically build on the distributionally robust optimization (DRO) framework, where the worst-case loss is minimized instead of the average loss (Ben-Tal et al., 2013; Hu et al., 2018; Sagawa et al., 2019; Oren et al., 2019; Zhang et al., 2020). Notably, Group DRO (Sagawa et al., 2019), which optimizes a soft version of the worst-group loss holds state-of-the-art results on multiple benchmarks with spurious correlations.

Several methods have been proposed for the scenario where group labels are not known, and need to be inferred from the data; these methods typically train a pair of networks, where the first model is used to identify the challenging minority examples and define a weighted loss for the second model (Liu et al., 2021; Nam et al., 2020; Yaghoobzadeh et al., 2019; Utama et al., 2020; Creager et al., 2021; Dagaev et al., 2021; Zhang et al., 2022). Other works proposed semi-supervised methods for the scenario where the group labels are provided for a small fraction of the train datapoints (Sohoni et al., 2021; Nam et al., 2022). Idrissi et al. (2021) recently showed that with careful tuning simple approaches such as data subsampling and reweighting can provide competitive performance. We note that all of the methods described above use a validation set with a high representation of minority groups to tune the hyper-parameters and optimize worst-group performance.

In a closely related work, Menon et al. (2020) considered classifier retraining and threshold correction on a subset of the training data for correcting the subgroup bias. In our work, we focus on classifier retraining and show that re-using train data for last layer retraining as done in Menon et al. (2020) is suboptimal, while retraining the last layer on held-out data achieves significantly better performance across various spurious correlations benchmarks (see Section 6). Moreover, we make multiple contributions beyond the scope of Menon et al. (2020), for example: we analyze feature learning in the extreme simplicity bias scenarios (Section 4), show strong performance on natural language processing datasets with spurious correlations (Section 6) and demonstrate how last layer retraining can be used to correct background and texture bias in large-scale models on ImageNet (Section 7).

In an independent and concurrent work, Rosenfeld et al. (2022) show that ERM learns high quality representations for domain generalization, and by training the last layer on the target domain it is possible to achieve strong results. Kang et al. (2019) propose to use classifier re-training, among other methods, in the context of long-tail classification. The observations in our work are complimentary, as we focus on spurious correlation robustness instead of domain generalization and long-tail classification. Moreover, there are important algorithmic differences between DFR, and the methods of Kang et al. (2019) and Rosenfeld et al. (2022). In particular, Kang et al. (2019) retrain the last layer on the training data with class-balanced data sampling, while we use held-out data and group-balanced subsampling; they also do not use regularization. Rosenfeld et al. (2022) only present last layer retraining results as motivation, and do not use it in their proposed DARE method. We present a detailed discussion of these differences in Appendix A, and empirical comparisons in Appendix G. Due to the page limit, we provide further discussion of prior work in spurious correlations, transfer learning and other related areas in Appendix A.

## 4 UNDERSTANDING REPRESENTATION LEARNING WITH SPURIOUS CORRELATIONS

In this section we investigate the solutions learned by standard ERM classifiers on datasets with spurious correlations. We show that while these classifiers underperform on the minority groups, they still learn the core features that can be used to make correct predictions on the minority groups.

### 4.1 FEATURE LEARNING ON WATERBIRDS DATA

We first consider the Waterbirds dataset (Sagawa et al., 2019) (see Section 2) which is generated synthetically by combining images of birds from the CUB dataset (Wah et al., 2011) and backgrounds from the Places dataset (Zhou et al., 2017) (see Sagawa et al. (2019) for a detailed description of the data generation process). For the experiments in this section, we generate several variations of the Waterbirds data following the procedure analogous to Sagawa et al. (2019). The *Original* dataset is analogous to the standard Waterbirds data, but we vary the degree of spurious correlation between the background and the target: 50% (*Balanced* dataset), 95% (as in Sagawa et al. (2019)) and 100% (no minority group examples in the

| Train Data | Test Data (Worst Acc) | |
|---|---|---|
| (Spurious Corr.) | Original | FG-Only |
| Balanced (50%) | 91.9% | 94.7% |
| Original (95%) | 73.8% | 93.7% |
| Original (100%) | 38.4% | 94% |
| FG-Only (-) | 75.2% | 95.5% |

Table 1: ERM classifiers trained on Waterbirds with Original and FG-Only images achieve similar FG-Only accuracy.

train dataset). The *FG-Only* dataset contains images of the birds on uniform grey background instead of the Places background, removing the spurious feature. We show examples of datapoints from each variation of the dataset in Appendix Figure 4. In Appendix B.1, we provide the full details for the

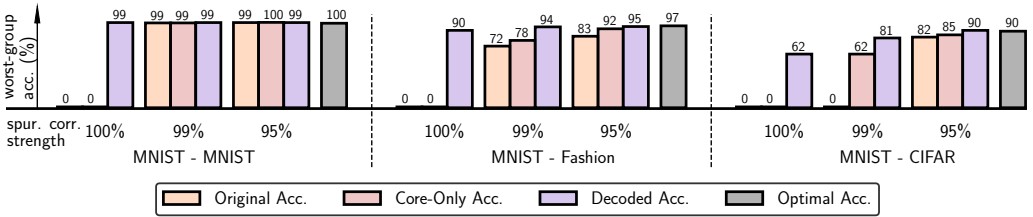

Figure 2: **Feature learning and simplicity bias.** ResNet-20 ERM classifiers trained on Dominoes data with varying levels of spurious correlation between core and spurious features. We show worst-group test accuracy for: Original data, data with only core features present (Core-Only), and accuracy of decoding the core feature from the latent representations of the Original data with logistic regression. We additionally report optimal accuracy: accuracy of a model trained and evaluated on the Core-Only data. Even in cases when the model achieves $0\%$ accuracy on the Original data, the core features can still be decoded from latent representations.

experiment in this section, and additionally consider the reverse Waterbirds problem, where instead of predicting the bird type, the task is to predict the background type, with similar results.

We train a ResNet-50 model with ERM on each of the variations of the data and report the results in Table 1. Following prior work (e.g., Sagawa et al., 2019; Liu et al., 2021; Idrissi et al., 2021), we initialize the model with weights pretrained on ImageNet. For the models trained on the Original data, there is a large difference between the mean and worst group accuracy on the Original test data: the model heavily relies on the background information in its predictions, so the performance on minority groups is poor. The model trained without minority groups is especially affected, only achieving $38.4\%$ worst group accuracy. However, surprisingly, we find that all the models trained on the Original data can make much better predictions on the FG-Only test data: if we remove the spurious feature (background) from the inputs at test time, the models make predictions based on the core feature (bird), and achieve worst-group[1] accuracy close to $94\%$, which is only slightly lower than the accuracy of a model trained directly on the FG-Only data and comparable to the accuracy of a model that was trained on balanced train data[2].

In summary, we conclude that while the models pre-trained on ImageNet and trained on the Original data make use of background information to make predictions, they still learn the features relevant to classifying the birds *almost as well as the models trained on the data without spurious correlations*. We will see that we can retrain the last layer of the Original-trained models and dramatically improve the worst-group performance on the Original data, by emphasizing the features relevant to the bird. While we use pre-trained models in this section, following the common practice on Waterbirds, we show similar results on other datasets without pre-training (see Section 4.2, Appendix C.1).

## 4.2 SIMPLICITY BIAS

Shah et al. (2020) showed that neural networks can suffer from *extreme simplicity bias*, a tendency to completely rely on the simple features, while ignoring similarly predictive (or even more predictive) complex features. In this section, we explore whether the neural networks can still learn the core features in the extreme simplicity bias scenarios, where the spurious features are simple and highly correlated with the target. Following Shah et al. (2020) and Pagliardini et al. (2022), we consider *Dominoes* binary classification datasets, where the top half of the image shows MNIST digits (LeCun et al., 1998) from classes $\{0, 1\}$, and the bottom half shows MNIST images from classes $\{7, 9\}$ (*MNIST-MNIST*), Fashion-MNIST (Xiao et al., 2017) images from classes {coat, dress} (*MNIST-Fashion*) or CIFAR-10 (Krizhevsky et al., 2009) images from classes {car, truck} (*MNIST-CIFAR*). In all Dominoes datasets, the top half of the image (MNIST $0 - 1$ images) presents a linearly separable feature; the bottom half of the image presents a harder to learn feature. See Appendix B.2 for more details about the experimental set-up and datasets, and Appendix Figure 4 for image examples.

---

[1]On the FG-Only data the groups only differ by the bird type, as we remove the background. The difference between mean and worst-group accuracy is because the target classes are not balanced in the training data.

[2]In Appendix B.4, we demonstrate *logit additivity*: the class logits on Waterbirds are well approximated as the sum of the logits for the corresponding background image and the logits for the foreground image.

We use the simple feature (top half of the images) as a spurious feature and the complex feature (bottom half) as the core feature; we generate datasets with $100\%$, $99\%$ and $95\%$ correlations between the spurious feature and the target in train data, while the core feature is perfectly aligned with the target. We then define $4$ groups $\mathcal{G}_i$ based on the values of the spurious and core features, where the minority groups correspond to images with top and bottom halves that do not match. The groups on validation and test are balanced.

We train a ResNet-20 model on each variation of the dataset. In Figure 2 we report the worst group performance for each of the datasets and each spurious correlation strength. In addition to the worst-group accuracy on the Original test data, we report the *Core-Only* worst-group accuracy, where we evaluate the model on datapoints with the spurious top half of the image replaced with a black image. Similarly to Shah et al. (2020) and Pagliardini et al. (2022), we observe that when the spurious features are perfectly correlated with the target on Dominoes datasets, the model relies just on the simple spurious feature to make predictions and achieves $0\%$ worst-group accuracy. However, with $99\%$ and $95\%$ spurious correlation levels on train, we observe that models learned the core features well, as indicated both by their performance on the Original test data and especially increased performance on Core-Only test data where spurious features are absent. For reference, on each dataset we also report the *Optimal accuracy*, which is the accuracy of a model trained and evaluated on the Core-Only data. The optimal accuracy provides an upper bound on the accuracy that we can expect on each of the datasets.

**Decoding feature representations.** The performance on the Original and even Core-Only data might not be fully representative of whether or not the network learned a high-quality representation of the core features. Indeed, even if we remove the MNIST digit from the top half of the image, the network can still primarily rely on the (empty) top half in its predictions: an empty image may be more likely to come from class 1, which typically has fewer white pixels than class 0. To see how much information about the core feature is contained in the latent representation, we evaluate the *decoded* accuracy: for each problem we train a logistic regression classifier on the features extracted by the final convolutional layer of the network. We use a group-balanced validation set to train the logistic regression model, and then report the worst-group accuracy on a test set. In Figure 2, we observe that for MNIST-MNIST and MNIST-FashionMNIST even when the spurious correlation is $100\%$, reweighting the features leads to high worst group test accuracy. Moreover, on all Dominoes datasets for $99\%$ and $95\%$ spurious correlations level the core features can be decoded with high accuracy and almost match the optimal accuracy. This decoding serves as a basis of our DFR method, which we describe in detail in Section 5. In Appendix B.2 we report an additional baseline and verify that the model indeed learns a non-trivial representation of the core feature.

**Relation to prior work.** With the same Dominoes datasets that we consider in this section, Shah et al. (2020) showed that neural networks tend to rely entirely on the simple features. However, they only considered the 100% spurious correlation strength and accuracy on the Original test data. Our results do not contradict their findings but provide new insights: even in these most challenging cases, the networks still represent information about the complex core feature. Moreover, this information can be decoded to achieve high accuracy on the mixed group examples. Hermann & Lampinen (2020) considered a different set of synthetic datasets, showing that in some cases neural networks fail to represent information about some of the predictive features. In particular, they also considered decoding the information about these features from the latent representations and different spurious correlation strengths. Our results add to their observations and show that while it is possible to construct examples where predictive features are suppressed, in many challenging practical scenarios, neural networks learn a high quality representation of the core features relevant to the problem even if they rely on the spurious features.

**ColorMNIST.** In Appendix B.3 we additionally show results on ColorMNIST dataset with varied spurious correlations strength in train data: decoding core features from the trained representations on this problem also achieves results close to optimal accuracy and demonstrates that the core features are learned even in the cases when the model initially achieved $0\%$ worst-group accuracy.

In summary, we find that, surprisingly, if (1) the strength of the spurious correlation is lower than $100\%$ or (2) the difference in complexity between the core and spurious features is not as stark as on MNIST-CIFAR, *the core feature can be decoded from the learned embeddings with high accuracy*.

## 5 DEEP FEATURE REWEIGHTING

In Section 4 we have seen that neural networks trained with standard ERM learn multiple features relevant to the predictive task, such as features of both the background and the object represented in the image. Inspired by these observations, we propose *Deep Feature Reweighting (DFR)*, a simple and practical method for improving robustness to spurious correlations and distribution shift.

Let us assume that we have access to a dataset $\mathcal{D} = \{x_i, y_i\}$ which can exhibit spurious correlations. Furthermore, we assume that we have access to a (typically much smaller) dataset $\hat{\mathcal{D}}$, where the groups are represented equally. $\hat{\mathcal{D}}$ can be a subset of the train dataset $\mathcal{D}$, or a separate set of datapoints. We will refer to $\hat{\mathcal{D}}$ as *reweighting dataset*. We start by training a neural network on all of the available data $\mathcal{D}$ with standard ERM without any group or class reweighting. For this stage we do not need any information beyond the training data and labels. Here, we assume that the network consists of a feature extractor (such as a sequence of convolutional or transformer layers), followed by a fully-connected classification layer mapping the features to class logits. In the second stage of the procedure, we simply discard the classification head and train a new classification head from scratch on the available balanced data $\hat{\mathcal{D}}$. We use the new classification head to make predictions on the test data. We illustrate DFR in Figure 1. **Notation:** we will use notation $\text{DFR}_{\mathcal{D}}^{\hat{\mathcal{D}}}$, where $\mathcal{D}$ is the dataset used to train the base feature extractor model and $\hat{\mathcal{D}}$ – to train the last linear layer.

## 6 FEATURE REWEIGHTING IMPROVES ROBUSTNESS

In this section, we evaluate DFR on benchmark problems with spurious correlations.

**Data.** See Section 2 for a description of the *Waterbirds* data. On *CelebA* hair color prediction(Liu et al., 2015), the groups are non-blond females ($\mathcal{G}_1$), blond females ($\mathcal{G}_2$), non-blond males ($\mathcal{G}_3$) and blond males ($\mathcal{G}_4$) with proportions $44\%, 14\%, 41\%$, and $1\%$ of the data, respectively; the group $\mathcal{G}_4$ is the minority group, and the gender serves as a spurious feature. On *MultiNLI*, the goal is to classify the relation between a pair of sentences (premise and hypothesis) as contradiction, entailment or neutral; the presence of negation words ("no", "never", ...) is correlated with the contradiction class, and serves as a spurious feature. Finally, we consider the *CivilComments* (Borkan et al., 2019) dataset implemented in the WILDS benchmark (Koh et al., 2021; Sagawa et al., 2021), where the goal is to classify comments as toxic or neutral. Each comment is labeled with $8$ attributes $s_i$ (male, female, LGBT, black, white, Christian, Muslim, other religion) based on whether or not the corresponding characteristic is mentioned in the comment. For evaluation, we follow the standard WILDS protocol and report the worst accuracy across the 16 overlapping groups corresponding to pairs $(y, s_i)$ for each of the attributes $s_i$. See Figures 6, 7 for a visual description of all four datasets.

**Baselines.** We consider 6 baseline methods that work under different assumptions on the information available at the training time. *Empirical Risk Minimization* (*ERM*) represents conventional training without any procedures for improving worst-group accuracies. *Just Train Twice* (*JTT*) (Liu et al., 2021) is a method that detects the minority group examples on train data, only using group labels on the validation set to tune hyper-parameters. *Correct-n-Contrast* (*CnC*) (Zhang et al., 2022) detects the minority group examples similarly to JTT, and uses a contrastive objective to learn representations robust to spurious correlations. *Group DRO* (Sagawa et al., 2019) is a state-of-the-art method, which uses group information on train and adaptively upweights worst-group examples during training. *SUBG* is ERM applied to a random subset of the data where the groups are equally represented, which was recently shown to be a surprisingly strong baseline (Idrissi et al., 2021). Finally, *Spread Spurious Attribute* (*SSA*) (Nam et al., 2022) attempts to fully exploit the group-labeled validation data with a semi-supervised approach that propagates the group labels to the the training data. We discuss the assumptions on the data for each of these baselines in Appendix C.2.

**DFR.** We evaluate $DFR_{Tr}^{Val}$, where we use a group-balanced[3] subset of the validation data available for each of the problems as the reweighting dataset $\hat{\mathcal{D}}$. We use the standard training dataset (with group imbalance) $\mathcal{D}$ to train the feature extractor. In order to make use of more of the available data, we train logistic regression 10 times using different random balanced subsets of the data, and average the weights of the learned models. We report full details and several ablation studies in Appendix C.

---

[3]We keep all of the data from the smallest group, and subsample the data from the other groups to the same size. On CivilComments the groups overlap, so for each datapoint we use the smallest group that contains it when constructing the group-balanced reweighting dataset.

| Method | Group Info | Waterbirds | | CelebA | | MultiNLI | | CivilComments | |
|---|---|---|---|---|---|---|---|---|---|
| | Train / Val | Worst(%) | Mean(%) | Worst(%) | Mean(%) | Worst(%) | Mean(%) | Worst(%) | Mean(%) |
| JTT | ✗ / ✓ | 86.7 | 93.3 | 81.1 | 88.0 | 72.6 | 78.6 | 69.3 | 91.1 |
| CnC | ✗ / ✓ | $88.5_{\pm0.3}$ | $90.9_{\pm0.1}$ | $88.8_{\pm0.9}$ | $89.9_{\pm0.5}$ | - | - | $68.9_{\pm2.1}$ | $81.7_{\pm0.5}$ |
| SUBG | ✓ / ✓ | $89.1_{\pm1.1}$ | - | $85.6_{\pm2.3}$ | - | $68.9_{\pm0.8}$ | - | - | - |
| SSA | ✗ / ✓✓ | $89.0_{\pm0.6}$ | $92.2_{\pm0.9}$ | $\mathbf{89.8}_{\pm1.3}$ | $92.8_{\pm0.1}$ | $76.6_{\pm0.7}$ | $79.9_{\pm0.87}$ | $\mathbf{69.9}_{\pm2}$ | $88.2_{\pm2.}$ |
| Group DRO | ✓ / ✓ | 91.4 | 93.5 | 88.9 | 92.9 | $\mathbf{77.7}$ | 81.4 | 69.9 | 88.9 |
| Base (ERM) | ✗ / ✗ | $74.9_{\pm2.4}$ | $98.1_{\pm0.1}$ | $46.9_{\pm2.8}$ | $95.3_{\pm0}$ | $65.9_{\pm0.3}$ | $82.8_{\pm0.1}$ | $55.6_{\pm0.6}$ | $92.1_{\pm0.1}$ |
| $\text{DFR}_{\text{Tr}}^{\text{Val}}$ | ✗ / ✓✓ | $\mathbf{92.9}_{\pm0.2}$ | $94.2_{\pm0.4}$ | $88.3_{\pm1.1}$ | $91.3_{\pm0.3}$ | $74.7_{\pm0.7}$ | $82.1_{\pm0.2}$ | $\mathbf{70.1}_{\pm0.8}$ | $87.2_{\pm0.3}$ |

Table 2: **Spurious correlation benchmark results.** Worst-group and mean test accuracy of DFR variations and baselines on benchmark datasets. For mean accuracy, we follow Sagawa et al. (2019) and weight the group accuracies according to their prevalence in the training data. The Group Info column shows whether group labels are available to the methods on train and validation datasets. $\text{DFR}_{\text{Tr}}^{\text{Val}}$ uses the validation data to train the model parameters (last layer) in addition to hyperparameter tuning, indicated with ✓✓; SSA also uses the validation set to train the model. For DFR we report the mean±std over 5 independent runs. DFR is competitive with state-of-the-art.

**Hyper-parameters.** Following prior work (e.g. Liu et al., 2021), we use a ResNet-50 model (He et al., 2016) pretrained on ImageNet for Waterbirds and CelebA and a BERT model (Devlin et al., 2018) pre-trained on Book Corpus and English Wikipedia data for MultiNLI and CivilComments. For $\text{DFR}_{\text{Tr}}^{\text{Val}}$, the size of the reweighting set $\hat{\mathcal{D}}$ is small relative to the number of features produced by the feature extractor. For this reason, we use $\ell_1$-regularization to allow the model to learn sparse solutions and drop irrelevant features. For $\text{DFR}_{\text{Tr}}^{\text{Val}}$ *we only tune a single hyper-parameter* — the strength of the regularization term. We tune all the hyper-parameters on the validation data provided with each of the datasets. We note that the prior work methods including Group DRO, JTT, CnC, SUBG, SSA and others extensively tune hyper-parameters on the validation data. For $\text{DFR}_{\text{Tr}}^{\text{Val}}$ we split the validation in half, and use one half to tune the regularization strength parameter; then, we retrain the logistic regression with the optimal regularization on the full validation set.

**Results.** We report the results for $\text{DFR}_{\text{Tr}}^{\text{Val}}$ and the baselines in Table 2. $\text{DFR}_{\text{Tr}}^{\text{Val}}$ is competitive with the state-of-the-art Group DRO across the board, achieving the best results among all methods on Waterbirds and CivilComments. Moreover, $\text{DFR}_{\text{Tr}}^{\text{Val}}$ achieves similar performance to SSA, a method designed to make optimal use of the group information on the validation data ($\text{DFR}_{\text{Tr}}^{\text{Val}}$ is slightly better on Waterbirds and CivilComments while SSA is better on CelebA and MultiNLI). Both methods use the same exact setting and group information to train and tune the model. We explore other variations of DFR in Appendix C.3.

To sum up, $\text{DFR}_{\text{Tr}}^{\text{Val}}$ matches the performance of the best available methods, while only using the group labels on a small validation set. This state-of-the-art performance is achieved by simply reweighting the features learned by standard ERM, with no need for advanced regularization techniques.

## 7 NATURAL SPURIOUS CORRELATIONS ON IMAGENET

In the previous section we focused on benchmark problems constructed to highlight the effect of spurious features. Computer vision classifiers are known to learn undesirable patterns and rely on spurious features in real-world problems (see e.g. Rosenfeld et al., 2018; Singla & Feizi, 2021; Geirhos et al., 2020). In this section we explore two prominent shortcomings of ImageNet classifiers: background reliance (Xiao et al., 2020) and texture bias (Geirhos et al., 2018).

### 7.1 BACKGROUND RELIANCE

Prior work has demonstrated that computer vision models such as ImageNet classifiers can rely on image background to make their predictions (Zhang et al., 2007; Ribeiro et al., 2016; Shetty et al., 2019; Xiao et al., 2020; Singla & Feizi, 2021). Here, we show that it is possible to reduce the background reliance of ImageNet-trained models by simply retraining their last layer with DFR.

Xiao et al. (2020) proposed several datasets in the *Backgrounds Challenge* to study the effect of the backgrounds on predictions of ImageNet models. The datasets in the Backgrounds Challenge are based on the ImageNet-9 dataset, a subset of ImageNet structured into 9 coarse-grain classes (see Xiao et al. (2020) for details). ImageNet-9 contains $45k$ training images and 4050 validation images. We consider three datasets from the Backgrounds Challenge: (1) *Original* contains the

Figure 3: **Background reliance.** Accuracy of DFR$^{MR}$ and DFR$^{OG+MR}$ on different ImageNet-9 validation splits with an ImageNet-trained ResNet-50 feature extractor. DFR reduces background reliance with a minimal drop in performance on the Original data.

original images; (2) *Mixed-Rand* contains images with random combinations of backgrounds and foregrounds (objects); (3) *FG-Only* contains images showing just the object with a black background. We additionally consider *Paintings-BG* using paintings from Kaggle's `painter-by-numbers` dataset ( `https://www.kaggle.com/c/painter-by-numbers/`) as background for the ImageNet-9 validation data. Finally, we consider the ImageNet-R dataset (Hendrycks et al., 2021) restricted to the ImageNet-9 classes. See Appendix D for details and Figure 11 for example images.

We use an ImageNet-trained ResNet-50 as a feature extractor and train DFR with different reweighting datasets. As a *Baseline*, we train DFR on the Original $45k$ training datapoints (we cannot use the ImageNet-trained ResNet-50 last layer, as ImageNet-9 has a different set of classes). We train *DFR$^{MR}$* and *DFR$^{OG+MR}$* on Mixed-Rand training data and a combination of Mixed-Rand and Original training data respectively. In Figure 3, we report the predictive accuracy of these methods on different validation datasets as a function of the number of Mixed-Rand data observed. We select the observed subset of the data randomly; for DFR$^{OG+MR}$, in each case we use the same amount of Mixed-Rand and Original training datapoints. We repeat this experiment with a VIT-B-16 model (Dosovitskiy et al., 2020) pretrained on ImageNet-21$k$ and report the results in the Appendix Figure 9.

First, we notice that the baseline model provides significantly better performance on FG-Only (92%) than on the Mixed-Rand (86%) validation set, suggesting that the feature extractor learned the features needed to classify the foreground, as well as the background features. With access to Mixed-Rand data, we can reweight the foreground and background features with DFR and significantly improve the performance on mixed-rand, FG-Only and Paintings-BG datasets. At the same time, DFR$^{OG+MR}$ is able to mostly maintain the performance on the Original ImageNet-9 data; the small drop in performance is because the background is relevant to predicting the class on this validation set. Finally, on ImageNet-R, DFR provides a small improvement when we use all of $45k$ datapoints; the covariate shift in ImageNet-R is not primarily background-based, so reducing background reliance does not provide a big improvement. We provide additional results in See Appendix D.

## 7.2 TEXTURE-VS-SHAPE BIAS

Geirhos et al. (2018) showed that on images with conflicting texture and shape, ImageNet-trained CNNs tend to make predictions based on the texture, while humans usually predict based on the shape of the object. The authors designed the *GST* dataset with conflicting cues, and proposed the term *texture bias* to refer to the fraction of datapoints on which a model (or a human) makes predictions based on texture; conversely, *shape bias* is the fraction of the datapoints on which prediction is made based on the shape of the object. Geirhos et al. (2018) showed that it is possible to increase the shape bias of CNNs by training on Stylized ImageNet (*SIN*), a dataset obtained from ImageNet by removing the texture information via style transfer (see Appendix Figure 11 for example images). Using SIN in combination with ImageNet (*SIN+IN*), they also obtained improved robustness to corruptions. Finally, they proposed the *Shape-RN-50* model, a ResNet-50 (RN-50) trained on SIN+IN and finetuned on IN, which outperforms the ImageNet-trained ResNet-50 on in-distribution data and out-of-distribution robustness.

Here, we explore whether it is possible to change the shape bias of ImageNet-trained models by simply retraining the last layer with DFR. Intuitively, we expect that the standard ImageNet-trained model already learns *both* the shape and texture features. Indeed, Hermann et al. (2020) showed that

| Method | Training Data | Shape bias (%) | Top-1 Acc (%) | | |
|---|---|---|---|---|---|
| | | | ImageNet | ImageNet-R | ImageNet-C |
| RN-50 | IN | 21.4 | 76.0 | 23.8 | 39.8 |
| | SIN | **81.4** | 60.3 | 26.9 | 38.1 |
| | IN+SIN | 34.7 | 74.6 | **27.6** | **45.7** |
| Shape-RN-50 | IN+SIN | 20.5 | **76.8** | 25.6 | 42.3 |
| DFR | SIN | 34.0 | 65.1 | 24.6 | 36.7 |
| | IN+SIN | 30.6 | 74.5 | **27.2** | 40.7 |

Table 3: **Shape bias.** Shape bias and accuracy on ImageNet validation set variations for ResNet-50 trained on different datasets and DFR with an ImageNet-trained ResNet-50 as a feature extractor. For each metric, we show the best result in bold. By retraining just the last layer with DFR, we can significantly increase the shape bias compared to the base model ($21.4\% \rightarrow 34\%$ for DFR(SIN)) and improve performance on ImageNet-R/C.

shape information is partially decodable from the features of ImageNet models on the GST dataset. Here, instead of targeting GST, we apply DFR to the large-scale SIN dataset, and explore both the shape bias and the predictive performance of the resulting models. In Table 3, we report the shape bias, as well as predictive accuracy of ResNet-50 models trained on ImageNet (*IN*), SIN, IN+SIN and the Shape-RN-50 model, and the DFR models trained on SIN and IN+SIN. The DFR models use an IN-trained ResNet-50 model as a feature extractor. See Appendix E for details.

First, we observe that while the SIN-trained RN-50 achieves a shape bias of 81.4%, as reported by Geirhos et al. (2018), the models trained on combinations of IN and SIN are still biased towards texture. Curiously, the Shape-RN-50 model proposed by Geirhos et al. (2018) has almost identical shape bias to a standard IN-trained RN-50! At the same time, Shape-RN-50 outperforms IN-trained RN-50 on all the datasets that we consider, and Geirhos et al. (2018) showed that Shape-RN-50 significantly outperforms IN-trained RN-50 in transfer learning to a segmentation problem, suggesting that it learned better shape-based features. The fact that the shape bias of this model is lower than that of an IN-trained RN-50, suggests that the shape bias is largely affected by the last linear layer of the model: even if the model extracted high-quality features capturing the shape information, the last layer can still assign a higher weight to the texture information.

Next, DFR can significantly increase the shape bias of an IN-trained model. $\text{DFR}_{\text{IN}}^{\text{SIN}}$ achieves a comparable shape bias to that of a model trained from scratch on a combination of IN and SIN datasets. Finally, $\text{DFR}_{\text{IN}}^{\text{IN+SIN}}$ improves the performance on both ImageNet-R and ImageNet-C (Hendrycks & Dietterich, 2019) datasets compared to the base RN-50 model. In the Appendix Table 11 we show similar results for a VIT-B-16 model pretrained on ImageNet-$21k$ and finetuned on ImageNet; there, $\text{DFR}_{\text{IN}}^{\text{IN+SIN}}$ can also improve the shape bias and performance on ImageNet-C, but does not help on ImageNet-R. To sum up, by reweighting the features learned by an ImageNet-trained model, we can significantly increase its shape bias and improve robustness to certain corruptions. However, to achieve the highest possible shape bias, it is still preferable to re-train the model from scratch, as RN-50(SIN) achieves a much higher shape bias compared to all other methods.

## 8 Discussion

We have shown that neural networks simultaneously learn multiple different features, including relevant semantic structure, even in the presence spurious correlations. By retraining the last layer of the network with DFR, we can significantly reduce the impact of spurious features and improve worst-group-performance of the models. In particular, DFR achieves state-of-the-art performance on spurious correlation benchmarks, and can reduce the reliance of ImageNet trained models on background and texture information. DFR is extremely simple, cheap and effective: it only has one hyper-parameter, and we can run it on ImageNet-scale data in a matter of minutes. We hope that our results will be useful to practitioners and will inspire further research in feature learning in the presence of spurious correlations. We provide further discussion in Appendix F.

ACKNOWLEDGMENTS

We would like to thank Micah Goldblum, Wanqian Yang, Marc Finzi, Wesley Maddox, Robin Schirrmeister, Yogesh Balaji, Timur Garipov and Vadim Bereznyuk for helpful comments. This research is supported by NSF CAREER IIS-2145492, NSF I-DISRE 193471, NIH R01DA048764-01A1, NSF IIS-1910266, NSF 1922658 NRT-HDR: FUTURE Foundations, Translation, and Responsibility for Data Science, NSF Award 1922658, Meta Core Data Science, Google AI Research, BigHat Biosciences, Capital One, and an Amazon Research Award. This work was supported in part through the NYU IT High Performance Computing resources, services, and staff expertise.

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

APPENDIX OUTLINE

This appendix is organized as follows. In Section A we provide additional related work discussion. In Section B we present details on the experiments on feature learning in the presence of spurious correlations. In Section C we provide details on the results for the spurious correlation benchmark datasets and perform ablation studies. We provide details on the experiments on the reliance of ImageNet-trained models on the background in Section D and on the texture-bias in Section E. In Section F we provide further discussion of the results presented in the paper and future work. In Section G we compare DFR to methods proposed in Kang et al. (2019) and other last layer retraining variations.

**Code references.** In addition to the experiment-specific packages that we discuss throughout the appendix, we used the following libraries and tools in this work: NumPy (Harris et al., 2020), SciPy (Virtanen et al., 2020), PyTorch (Paszke et al., 2017), Jupyter notebooks (Kluyver et al., 2016), Matplotlib (Hunter, 2007), Pandas (McKinney, 2010), transformers (Wolf et al., 2019).

## A ADDITIONAL RELATED WORK

**Differences with Kang et al. (2019).** Our work and Kang et al. (2019) consider different settings: Kang et al. (2019) considers long-tail classification with class imbalance, while we consider spurious correlations and shortcut learning. In spurious correlation robustness, there is often no class imbalance, and the methods of Kang et al. (2019) cannot be directly applied. Our conceptual results, such as the ability to control the reliance on background or texture features in trained models are also orthogonal to the observations of Kang et al. (2019).

Algorithmically, DFR is related to the methods of Kang et al. (2019), but there are still important differences. In the Learning Weight Scaling (LWS) method, the authors rescale the logits of the classifier with scalar weights $f_i$: the weight of the $i$-th row of the weight matrix in the last layer is updated to $\hat{w}_i = f_i w_i$. The parameters $f_i$ are trained by minimizing the loss on the training set with class-balanced sampling. In particular, LWS is not full last layer retraining, as only one parameter per class is learned.

Kang et al. (2019) also proposed a Classifier Re-training (cRT) approach which retrains the last layer on the training set with class-balanced sampling, where we are equally likely to sample datapoints from each class. cRT is closer to DFR than LWS, as it retrains all parameters in the last layer.

The algorithmic differences of these methods with DFR are as follows:

- In DFR, we use a *group*-balanced subset of the data instead of *class*-balanced sampling. This distinction is important, as in the group robustness setting the classes are often balanced, so LWS and cRT are not immediately applicable to the spurious correlation setting.

- In DFR, we *subsample* the reweighting dataset to be group balanced instead of using class- or group-balanced *sampling*. Specifically, we produce a dataset where the number of examples in each group is the same, and only use these datapoints. This detail is hugely important, as group-balanced sampling does not produce classifiers robust to spurious correlations (e.g. see RWG and SUBG methods in Idrissi et al., 2021).

- In $\mathrm{DFR}_{\mathrm{Tr}}^{\mathrm{Val}}$ , we use *held-out data* for retraining the last layer, and not the training data. This is the important distinction between $\mathrm{DFR}_{\mathrm{Tr}}^{\mathrm{Val}}$ and $\mathrm{DFR}_{\mathrm{Tr}}^{\mathrm{Tr}}$ . LWS and cRT both retrain the classifier on the training data.

- There are also technical differences in how we train the last layer in DFR: we average the weights of several independent runs of last layer retraining on different group-balanced subsets of the data, and use strong $\ell_1$ regularization.

To sum up, both cRT and LWS are not immediately applicable to spurious correlation robustness. Moreover, even if we adapt cRT to the spurious correlation setting, there are still major differences with DFR: subsampling vs group-balanced sampling, held-out data vs training data, and regularization. We present comparisons to LWS, cRT and related last layer retraining variations on Waterbirds and CelebA in Appendix G, where we show that both DFR versions outperform methods proposed in Kang et al. (2019) and other ablations. In Appendix C, we also show that regularization and multiple re-runs of last layer retraining are important for strong performance with DFR.

**Differences with Rosenfeld et al. (2022).** While our works make similar high-level observations, they are actually complementary to each other. In particular, our works don't share any datasets, experiments or problem settings. Rosenfeld et al. (2022) focus on domain generalization, where the goal is to train a model that generalizes to unseen domains. In this setting, we know the domain labels on the train, and we have no data from the test domains. In spurious correlations, the goal is to train a model that does not rely on spurious features. We typically have examples from all the test groups in train, but the groups are highly imbalanced. As in domain generalization we do not have access to target domain data, Rosenfeld et al. (2022) refer to last layer retraining on the target domain as "cheating" (see e.g. the caption of Figure 1 in their paper). Consequently, they propose a different method, DARE, which is very different from DFR. DARE estimates means and covariance matrices for each domain to whiten out the features. On test, they apply approximate whitening to a new domain, and they don't use test domain data to retrain the last layer. On the other hand, DFR uses an ERM-trained feature extractor and simply retrains the last layer on a group-balanced reweighting dataset.

To sum up, the observations from our work and the concurrent work by Rosenfeld et al. (2022) are complimentary: we both show that ERM learns the core features well, but the settings, methods and experiments are different. In particular, it is not trivial to apply DARE to spurious correlations or DFR to domain generalization.

**Other group robustness methods.** Another group of papers proposes regularization techniques to learn diverse solutions on the train data, focusing on different groups of features (Teney et al., 2021a; Lee et al., 2022; Pagliardini et al., 2022; Pezeshki et al., 2021). Xu et al. (2022) show how to train *orthogonal classifiers*, i.e. classifiers invariant to given spurious features in the data. Other papers proposed methods based on meta-learning the weights for a weighted loss (Ren et al., 2018) and group-agnostic adaptive regularization (Cao et al., 2019; 2020). Sohoni et al. (2020) use clustering of internal features to provide approximate labels for group distributionally robust optimization. A number of prior works address de-biasing classifiers by leveraging prior knowledge on the bias type (Li & Vasconcelos, 2019; Kim et al., 2019; Tartaglione et al., 2021; Teney et al., 2021b; Zhu et al., 2021; Wang et al., 2019; Cadene et al., 2019; Li et al., 2018b).

**Spurious correlations in natural image datasets.** Multiple works demonstrated that natural image datasets contain spurious correlations that hurt neural network models (Kolesnikov & Lampert, 2016; Xiao et al., 2020; Shetty et al., 2019; Alcorn et al., 2019; Singla & Feizi, 2021; Singla et al., 2021; Moayeri et al., 2022). Notably, Geirhos et al. (2018) demonstrated that ImageNet-trained CNNs are biased towards texture rather than shape of the objects. Follow-up work explored this texture bias and showed that despite being texture-biased, CNNs still often represent information about the shape in their feature representations (Hermann et al., 2020; Islam et al., 2021). In Section 7 we show that it is possible to reduce the reliance of ImageNet-trained models on background context and texture information by retraining just the last layer of the model.

**Transfer learning.** Transfer learning (Pan & Yang, 2009; Sharif Razavian et al., 2014) is an extremely popular framework in modern machine learning. Multiple works demonstrate its effectiveness (e.g. Girshick, 2015; Huh et al., 2016; He et al., 2017; Sun et al., 2017; Mahajan et al., 2018; Kolesnikov et al., 2020; Maddox et al., 2021), and study when and why transfer learning can be effective (Zhai et al., 2019; Neyshabur et al., 2020; Abnar et al., 2021; Kornblith et al., 2019; Kumar et al., 2022). Bommasani et al. (2021) provide a comprehensive discussion of modern transfer learning with large-scale models. While algorithmically our proposed DFR method is related to transfer learning, it has a different motivation and works on different problems. We discuss the relation between DFR and transfer learning in detail in Section 5.

Related methods have been developed in several areas of machine learning, such as ML Fairness (Dwork et al., 2012; Hardt et al., 2016; Kleinberg et al., 2016; Pleiss et al., 2017; Agarwal et al., 2018; Khani et al., 2019), NLP (McCoy et al., 2019; Lovering et al., 2020; Kaushik et al., 2020; 2021; Eisenstein, 2022; Veitch et al., 2021) domain adaptation (Ganin & Lempitsky, 2015; Ganin et al., 2016) and domain generalization (Blanchard et al., 2011; Muandet et al., 2013; Li et al., 2018a; Gulrajani & Lopez-Paz, 2020; Ruan et al., 2021) including works on Invariant Risk Minimization and causality (Peters et al., 2016; Arjovsky et al., 2019; Krueger et al., 2021; Aubin et al., 2021).

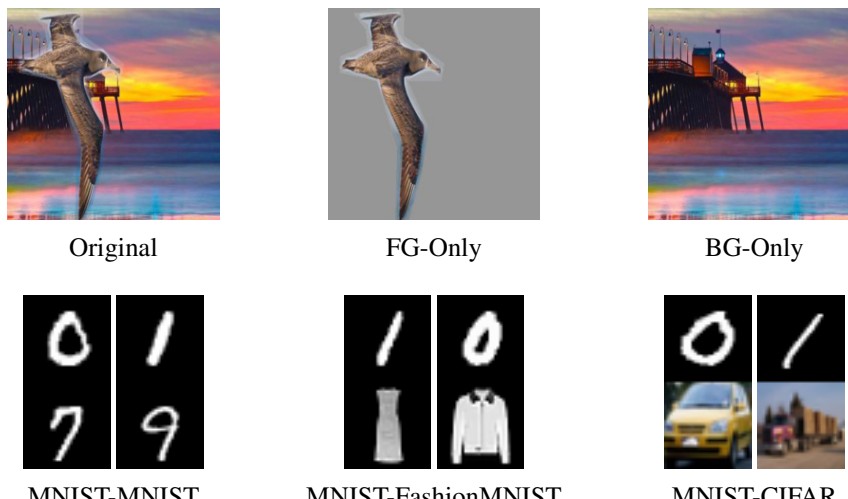

Figure 4: **Data examples.** Variations of the waterbirds (Top) and Dominoes (Bottom) datasets generated for the experiment in Section 4.

# B DETAILS: UNDERSTANDING REPRESENTATION LEARNING WITH SPURIOUS CORRELATIONS

Here we provide details on the experiments in Section 4.

## B.1 FEATURE LEARNING ON WATERBIRDS

**Inverse problem.** In Table 4 we present the results on the inverted Waterbirds problem, where the goal is to predict the background type while the bird type serves as a spurious feature. We note that prior work did not consider this reverse Waterbirds problem. The results for the inverted problem are analogous to the results for the standard Waterbirds (Table 1): models trained on the Original data learn the background features sufficiently well to predict the background type with high accuracy when the spurious foreground feature is not present, but perform poorly when presented with conflicting background and foreground features. While it is often suggested than neural networks are biased to learn the background (Xiao et al., 2020), we see that in fact the network relies on the spurious foreground feature (bird) when trained to predict the background.

**Data.** We show examples of Original, FG-Only and BG-Only Waterbirds images in Figure 4. To generate the data, we follow the instructions at `github.com/kohpangwei/group_DRO#waterbirds`, but in addition to the Original data we save the backgrounds and foregrounds separately. Consequently, the FG-Only data contains the same exact birds images as the Original data, and the BG-Only data contains the same exact places images as the Original data. For the 100% spurious correlation strength we simply discard all the minority groups data from the Original (95% spurious correlation) training dataset. For the Balanced data, we start with the Original data with 95% spurious correlation and replaced the background in the smallest possible number of images (chosen randomly) to achieve a 50% spurious correlation strength.

**Hyper-parameters.** For the experiments in this section we use a ResNet-50 model pretrained on ImageNet, imported from the `torchvision` package: `torchvision.models.resnet50(pretrained=True)` (Marcel & Rodriguez, 2010). We train the models for 50 epochs with SGD with a constant learning rate of $10^{-3}$, momentum decay of 0.9, batch size 32 and a weight decay of $10^{-2}$. We use random crops (`RandomResizedCrop(224, scale=(0.7, 1.0), ratio=(0.75, 4./3.), interpolation=2)`) and horizontal flips (`RandomHorizontalFlip()`) implemented in `torchvision.transforms` as data augmentation.

## B.2 DOMINOES DATASETS

**Data.** We show data examples from Dominoes datasets (MNIST-MNIST, MNIST-FashionMNIST and MNIST-CIFAR) in Figure 4. The top half of each image shows MNIST digits from classes

| Train Data | Spurious Corr. (%) | Test Data (Worst/Mean, %) | |
| --- | --- | --- | --- |
| | | Original | BG-Only |
| Balanced | 50 | 93.2/95.6 | 93.6/96.0 |
| Original | 95 | 77.4/91.2 | 93.1/95.7 |
| Original | 100 | 36.1/77.5 | 92.7/94.8 |
| Place-Only | - | 91.8/94.2 | 92.4/95.2 |

Table 4: **Feature learning on Inverted Waterbirds.** ERM classifiers trained on Inverted Waterbirds with Original and BG-Only images. Here the target is associated with the background type, and the foreground (bird type) serves as the spurious feature. All the models trained on the Original data including the model trained without any minority group examples (*Spurious corr.* 100%) underperform on the worst-group accuracy on the Original data, but perform well on the BG-Only data, almost matching the performance of the BG-Only trained model.

{0, 1}, and the bottom half shows: MNIST images from classes {7, 9} for MNIST-MNIST, Fashion-MNIST images from classes {coat, dress} for MNIST-FashionMNIST, and CIFAR-10 images from classes {car, truck} for MNIST-CIFAR. The label corresponds to the more complex bottom part of the image, but the top and bottom parts are correlated (we consider 95%, 99% and 100% levels of spurious correlation strength in experiments). 20% of the training data was reserved for validation or *reweighting* dataset (see Section 5) where each group is equally represented.

**Hyper-parameters.** We used a randomly initialized ResNet-20 architecture for this set of experiments. We trained the network for 500 epochs with SGD with batch size 32, weight decay $10^{-3}$, initial learning rate value $10^{-2}$ and a cosine annealing learning rate schedule. For the logistic regression model, we first extract the embeddings from the penultimate layer of the network, then use the logistic regression implementation (`sklearn.linear_model.LogisticRegression`) from the `scikit-learn` package (Pedregosa et al., 2011). We use $\ell_1$ regularization and tune the inverse regularization strength parameter $C$ in the range $\{0.1, 10, 100, 1000\}$. For more details on Deep Feature Reweighting implementation and tuning, see section C.

**Transfer from simple to complex features.** In addition to the results presented in Figure 2, we measure the decoded accuracy using the model trained just on the spurious simple features: the top half of the image showing an MNIST digit. After training, we retrain the last layer of the model using a validation split of the corresponding Dominoes dataset which has both top and bottom parts. In this case, we measure the transfer learning performance with the features learned on the binary MNIST classification problem applied to the more complex bottom half of the image. We obtain the following *transfer accuracy* results: 92.5% on MNIST-MNIST, 92.1% on MNIST-Fashion and 61.4% on MNIST-CIFAR. On all datasets, the decoded accuracy reported in Figure 2 for the spurious correlation levels 99% and 95% is better than the transfer accuracy. For the 100% spurious correlation, transfer achieves comparable results on MNIST-Fashion and MNIST-CIFAR, but on MNIST-MNIST the decoded accuracy with a model trained on the data with the core feature is significantly higher (99% vs 92%). These results confirm that for the spurious correlation strength below 100%, the model learns a high quality representation of the core features, which cannot be explained by transfer learning from the spurious feature.

### B.3 COLORMNIST

**Data.** We follow Zhang et al. (2022) to generate ColorMNIST dataset: there are 5 classes corresponding to pairs of digits $(0, 1), (2, 3), (4, 5), (6, 7), (8, 9)$, and 5 fixed colors such that in train data each class $y$ has a color $s$ which is associated with it with correlation $p_{corr}$, and the remaining examples are colored with the remaining colors uniformly at random. Validation split is a randomly sampled 10% fraction of the original MNIST data, and validation and test data exmaples are colored with 5 colors uniformly at random.

**Hyper-parameters.** We train a small LeNet-like Convolutional Neural Network model on this dataset varying $p_{corr}$. The model is trained for 5 epochs, with batch size 32, learning rate $10^{-3}$, and weight decay $5 \times 10^{-4}$, following Zhang et al. (2022). CNN consists of 2 convolutional layers with 6 and 16 output filters, followed by a max pooling layer, then another convolutional layer with 32 output filters and a fully-connected layer.

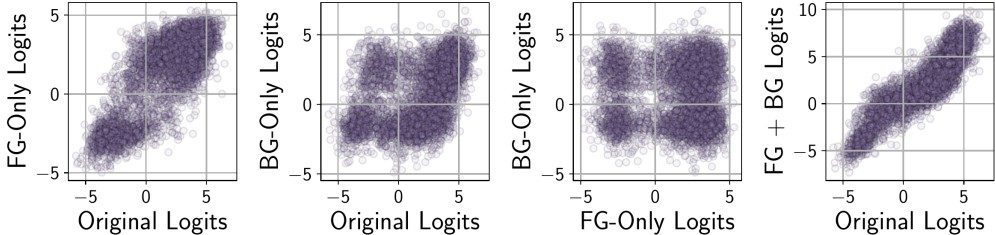

Figure 5: **Logit additivity.** Distribution of logits for the negative class on the test data for a model trained on Waterbirds dataset. We show scatter plots of the logits for the Original, FG-Only and BG-Only images. The logits on Original images are well aligned with sums of logits on the corresponding FG and BG images (rightmost panel), suggesting that the foreground and background features are processed close to independently in the network.

| $p_{corr}$ | no corr. | 0.8 | 0.9 | 0.95 | 0.995 | 1.0 |
|---|---|---|---|---|---|---|
| ERM | $94.5_{\pm 1.0}$ | $85.1_{\pm 2.2}$ | $66.0_{\pm 7.2}$ | $40.8_{\pm 3.6}$ | $0.0_{\pm 0.0}$ | $0.0_{\pm 0.0}$ |
| $\text{DFR}_{\text{Tr}}^{\text{Val}}$ | – | $93.8_{\pm 0.9}$ | $92.6_{\pm 0.7}$ | $91.6_{\pm 0.8}$ | $80.4_{\pm 1.1}$ | $77.3_{\pm 1.3}$ |

Table 5: **Results on ColorMNIST.** Worst-group accuracy of a small CNN model trained on 5-class ColorMNIST dataset, varying the spurious correlation strength $p_{corr}$ between classes and colors. DFR achieves strong performance even in the challenging settings with strong correlation between class labels and colors. We report mean $\pm$ std over 3 independent runs of the method.

We report worst-group accuracy for ERM and $\text{DFR}_{\text{Tr}}^{\text{Val}}$ in Table 5. The *no corr.* results for ERM correspond to the model trained on the dataset without correlation between colors and class labels. Notably, DFR recovers strong worst-group performance even in the cases when base model has 0% worst-group accuracy ($p_{corr} = 1.0$ and $p_{corr} = 0.995$). For $p_{corr} < 0.95$ DFR's worst-group accuracy is closely matching the optimal accuracy of the model trained on data without spurious correlations.

### B.4   LOGIT ADDITIVITY

To better understand why the models trained on the Original Waterbirds data perform well on FG-Only images, in Figure 5 we inspect the logits of a trained model. We show scatter plots of logits for the negative class (logits for the positive class behave analogously) on the Original, FG-Only and BG-Only test data. Both logits on FG-Only and BG-Only data correlate with the logits on the Original images, and FG-Only show a higher correlation. The FG-Only and BG-Only logits are not correlated with each other, as in test data the groups are balanced and the foreground and background are independent.

We find that the sum of the logits for the BG-Only and the logits for the FG-Only images provides a good approximation of the logits on the corresponding Original image (combining the foreground and the background). We term this phenomenon *logit additivity*: on Waterbirds, logits for the different predictive features (both core and spurious) are computed close to independently and added together in the last classification layer.

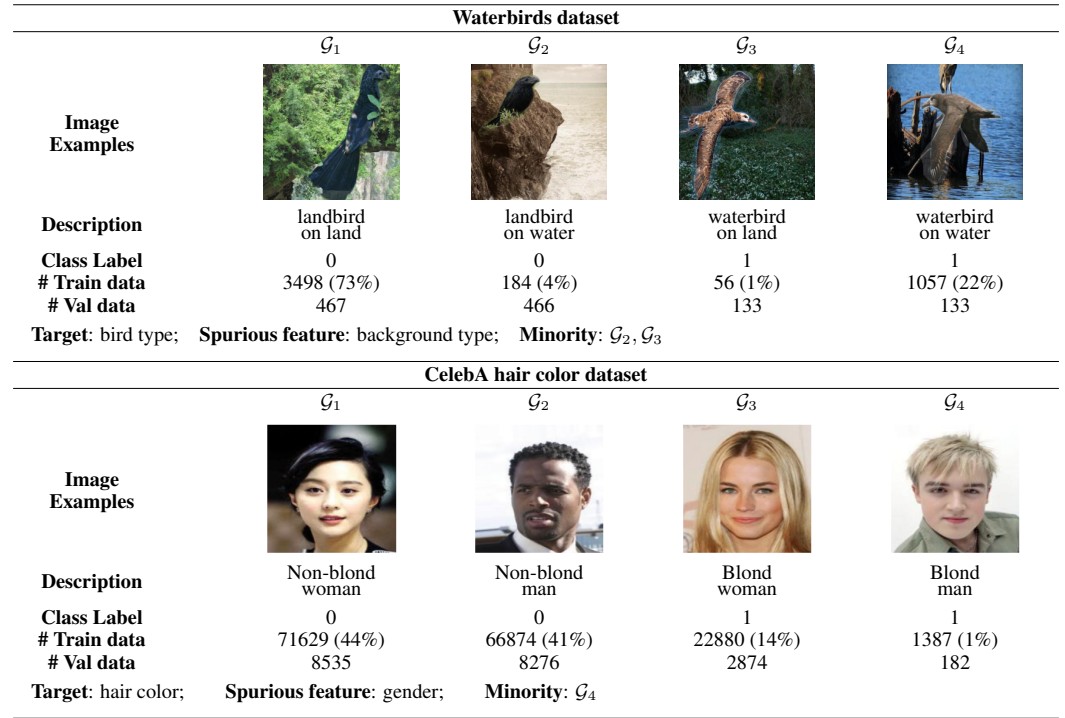

| Waterbirds dataset | | | | |
|---|---|---|---|---|
| | $\mathcal{G}_1$ | $\mathcal{G}_2$ | $\mathcal{G}_3$ | $\mathcal{G}_4$ |
| **Image Examples** | | | | |
| **Description** | landbird on land | landbird on water | waterbird on land | waterbird on water |
| **Class Label** | 0 | 0 | 1 | 1 |
| **# Train data** | 3498 (73%) | 184 (4%) | 56 (1%) | 1057 (22%) |
| **# Val data** | 467 | 466 | 133 | 133 |

**Target**: bird type;   **Spurious feature**: background type;   **Minority**: $\mathcal{G}_2, \mathcal{G}_3$

| CelebA hair color dataset | | | | |
|---|---|---|---|---|
| | $\mathcal{G}_1$ | $\mathcal{G}_2$ | $\mathcal{G}_3$ | $\mathcal{G}_4$ |
| **Image Examples** | | | | |
| **Description** | Non-blond woman | Non-blond man | Blond woman | Blond man |
| **Class Label** | 0 | 0 | 1 | 1 |
| **# Train data** | 71629 (44%) | 66874 (41%) | 22880 (14%) | 1387 (1%) |
| **# Val data** | 8535 | 8276 | 2874 | 182 |

**Target**: hair color;   **Spurious feature**: gender;   **Minority**: $\mathcal{G}_4$

Figure 6: **Waterbirds and CelebA data.** Dataset descriptions and example images from each group on Waterbirds and CelebA datasets.

## C   DETAILS: SPURIOUS CORRELATION BENCHMARKS

Here we provide details and additional ablations on the experiments in Section 6.

**Data.**   We use the standard Waterbirds, CelebA and MultiNLI datasets, following e.g. (Sagawa et al., 2019; Liu et al., 2021; Idrissi et al., 2021). We use CivilComments dataset as implemented in the WILDS benchmark (Koh et al., 2021). See Figures 6, 7 for group descriptions and data examples.

**Base model hyper-parameters.**   For the experiments on Waterbirds and CelebA we use a ResNet-50 model pretrained on ImageNet, imported from the `torchvision` package: `torchvision.models.resnet50(pretrained=True)`. We use random crops (`RandomResizedCrop(224, scale=(0.7, 1.0), ratio=(0.75, 4./3.), interpolation=2)`) and horizontal flips (`RandomHorizontalFlip()`) implemented in `torchvision.transforms` as data augmentation. We train all models with SGD with momentum decay of 0.9 and a constant learning rate. On Waterbirds, we train the models for 100 epochs with weight decay $10^{-3}$, learning rate $10^{-3}$ and batch size 32. On CelebA, we train the models for 50 epochs with weight decay $10^{-4}$, learning rate $10^{-3}$ and batch size 128. We do not use early stopping. On MultiNLI and Civil-Comments, we use the BERT for classification model from the `transformers` package: `BertForSequenceClassification.from_pretrained('bert-base-uncased', num_labels=num_classes)`. We train the BERT models with the AdamW (Loshchilov & Hutter, 2017) with linear learning rate annealing with initial learning rate $10^{-5}$, batch size 16 and weight decay $10^{-4}$ for 5 epochs.

**DFR details.**   For DFR, we first extract and save the embeddings (inputs to the classification layer of the base model) of the training, validation and testing data using the base model. We then preprocess the embeddings to have zero mean and unit standard deviation using the standard scaler `sklearn.preprocessing.StandardScaler` from the `scikit-learn` package. In each case, we compute the preprocessing statistics on the reweighting data used to train the last layer. To retrain the last layer, we use the logistic regression implementation (`sklearn.linear_model.LogisticRegression`) from the `scikit-learn` package. We use $\ell_1$ regularization. For $\text{DFR}_{\text{Tr}}^{\text{Val}}$ we only tune the inverse regularization strength parame-

| MultiNLI | | | | | |
|---|---|---|---|---|---|
| | **Text examples** | **Class label** | **Description** | **# Train data** | **# Val data** |
| $\mathcal{G}_1$ | *"if residents are unhappy, they can put wheels on their homes and go someplace else, she said. [SEP] residents are stuck here but they can't go anywhere else."* | 0 | contradiction no negations | 57498 (28%) | 22814 |
| $\mathcal{G}_2$ | *"within this conflict of values is a clash about art. [SEP] there is no clash about art."* | 0 | contradiction has negations | 11158 (5%) | 4634 |
| $\mathcal{G}_3$ | *"there was something like amusement in the old man's voice. [SEP] the old man showed amusement."* | 1 | entailment no negations | 67376 (32%) | 26949 |
| $\mathcal{G}_4$ | *"in 1988, the total cost for the postal service was about $36. [SEP] the postal service cost us citizens almost nothing in the late 80's. "* | 1 | entailment has negations | 1521 (1%) | 613 |
| $\mathcal{G}_5$ | *"yeah but even even cooking over an open fire is a little more fun isn't it [SEP] i like the flavour of the food."* | 2 | neutral no negations | 66630 (32%) | 26655 |
| $\mathcal{G}_6$ | *"that's not too bad [SEP] it's better than nothing"* | 2 | neutral has negations | 1992 (1%) | 797 |

**Target**: contradiction / entailment / neutral;    **Spurious feature**: has negation words.    **Minority**: $\mathcal{G}_4$, $\mathcal{G}_6$

| CivilComments | | | | | |
|---|---|---|---|---|---|
| | **Text examples** | **Class label** | **Description** | **# Train data** | **# Val data** |
| | *"I'm quite surprised this worked for you. Infrared rays cannot penetrate tinfoil."* | 0 | non-toxic no identities | 148186 (55%) | 25159 |
| | *"I think you may have misunderstood what 'straw men' are. But I'm glad that your gravy is good."* | 0 | non-toxic has identities | 90337 (33%) | 14966 |
| | *"Hahahaha putting his faith in Snopes. Pathetic."* | 1 | toxic no identities | 12731 (5%) | 2111 |
| | *"That sounds like something a white person would say."'* | 1 | toxic has identities | 17784 (7%) | 2944 |

**Target**: Toxic / not toxic comment;    **Spurious feature**: mentions protected categories.

Figure 7: **MultiNLI and CivilComments data.** Dataset descriptions and data examples from different groups on MultiNLI and CivilComments. We underline the words corresponding to the spurious feature. CivilComments contains 16 overlapping groups (corresponding to toxic / non-toxic comments and mentions of one of the protected identities: male, female, LGBT, black, white, Christian, Muslim, other religion. We only show examples with mentions of the male and white identities.

| Method | ImageNet Pretrain | Dataset Fine-tune | Waterbirds | | CelebA | |
|---|---|---|---|---|---|---|
| | | | Worst(%) | Mean(%) | Worst(%) | Mean(%) |
| Base Model | ✓ | ✓ | $74.9_{\pm 2.4}$ | $98.1_{\pm 0.1}$ | $46.9_{\pm 2.8}$ | $95.3_{\pm 0}$ |
| $\text{DFR}_{\text{Tr}}^{\text{Tr}}$ | ✓ | ✓ | $90.2_{\pm 0.8}$ | $97.0_{\pm 0.3}$ | $80.7_{\pm 2.4}$ | $85.4_{\pm 0.4}$ |
| $\text{DFR}_{\text{Tr}}^{\text{Val}}$ | ✓ | ✓ | $92.9_{\pm 0.2}$ | $94.2_{\pm 0.4}$ | $88.3_{\pm 1.1}$ | $89.6_{\pm 0.4}$ |
| Base Model | ✗ | ✓ | $6.9_{\pm 3.0}$ | $88.0_{\pm 1.1}$ | $39.8_{\pm 2.0}$ | $95.7_{\pm 0.1}$ |
| $\text{DFR}_{\text{Tr}}^{\text{Tr}}$ | ✗ | ✓ | $45.4_{\pm 4.1}$ | $69.8_{\pm 7.0}$ | $83.4_{\pm 2.6}$ | $87.5_{\pm 0.3}$ |
| $\text{DFR}_{\text{Tr}}^{\text{Val}}$ | ✗ | ✓ | $53.9_{\pm 1.8}$ | $62.6_{\pm 2.2}$ | $85.0_{\pm 2.1}$ | $87.6_{\pm 0.3}$ |
| $\text{DFR}_{\text{IN}}^{\text{Val}}$ | ✓ | ✗ | $88.7_{\pm 0.4}$ | $89.7_{\pm 0.1}$ | $73.1_{\pm 2.6}$ | $80.9_{\pm 0.5}$ |

Table 6: **Effect of ImageNet pretraining and dataset fine-tuning.** Results for DFR on the Waterbirds and CelebA datasets when using an ImageNet-trained model as a feature extractor, training the feature extractor from random initialization or initializing the feature extractor with ImageNet-trained weights and fine-tuning on the target data. On Waterbirds, we can achieve surprisingly strong worst group accuracy of $88.7\%$ without finetuning on the target data. On CelebA, we can achieve reasonable worst group accuracy of $85\%$ without pretraining. However, on both datasets, both ImageNet pretraining and dataset finetuning are needed to achieve optimal performance. All methods in Table 2 use ImageNet-trained models as initialization and finetune on the target dataset.

| Finetuned | WB | CelebA |
|---|---|---|
| Last Layer | $93.1_{\pm 0.2}$ | $88.3_{\pm 0.5}$ |
| Last 2 Layers | $93.1_{\pm 0.5}$ | $87.2_{\pm 0.9}$ |
| Last Block | $90.7_{\pm 0.4}$ | $83.0_{\pm 0.9}$ |
| All Layers | $35.6_{\pm 0.1}$ | $77.8_{\pm 0.1}$ |

Table 7: **Retraining multiple layers.** We retrain the last linear layer, last two layers (linear, last batchnorm and conv), last residual block, and all layers from scratch on group-balanced validation data. Retraining the last layer is sufficient for optimal performance, and moreover retraining more layers can hurt the results.

ter $C$: we consider the values in range $\{1., 0.7, 0.3, 0.1, 0.07, 0.03, 0.01\}$ and select the value that leads to the best worst-group performance on the available validation data (as described in Section 6, for $\text{DFR}_{\text{Tr}}^{\text{Val}}$ we use half of the validation to train the logistic regression model and the other half to tune the parameters at the tuning stage). For $\text{DFR}_{\text{Tr}}^{\text{Tr}}$ we additionally tune the class weights: we set the weight for one of the classes to $1$ and consider the weights for the other class in range $\{1, 2, 3, 10, 100, 300, 1000\}$; we then switch the classes and repeat the procedure. For the final evaluation, we use the best values of the hyper-parameters obtained during the tuning phase and train a logistic regression model on all of the available reweighting data. We train the logistic regression model on the reweighting data 10 times with random subsets of the data (we take all of the data from the smallest group, and subsample the other groups randomly to have the same number of datapoints) and average the weights of the learned models. We report the model with averaged weights.

## C.1 ABLATION STUDIES

**Is ImageNet pretraining necessary for image benchmarks?** DFR relies on the ability of the feature extractor to learn diverse features, which may suggest that ImageNet pretraining is crucial. In Appendix Table 6, we report the results on the same problems, but training the feature extractor from scratch. We find that on Waterbirds, ImageNet pretraining indeed has a dramatic effect on the performance of DFR as well as the base feature extractor model. However, on CelebA DFR shows strong performance regardless of pretraining. The difference between Waterbirds and CelebA is that Waterbirds contains only $4.8k$ training points, making it difficult to learn a meaningful feature extractor from scratch. Furthermore, on both datasets, finetuning the feature extractor on the target data is crucial: just using the features extracted by an ImageNet-trained model leads to poor results. We note that all the baselines considered in Table 2 use ImageNet pretraining.

**Retraining multiple layers.** In Table 7 we perform another ablation by retraining multiple layers from scratch on the group-balanced validation dataset on Waterbirds and CelebA. We find that retraining just the last layer provides optimal performance, retraining the last two layers (the last

| Base Model Hypers | | | | Method | Waterbirds | |
|---|---|---|---|---|---|---|
| lr | wd | batch size | aug | | Worst(%) | Mean(%) |
| $10^{-3}$ | $10^{-3}$ | 32 | ✓ | Base Model | $74.9_{\pm2.4}$ | $98.1_{\pm0.1}$ |
| | | | | $\text{DFR}_{\text{Tr}}^{\text{Val}}$ | $92.9_{\pm0.2}$ | $94.2_{\pm0.4}$ |
| $10^{-3}$ | $10^{-3}$ | 32 | ✗ | Base Model | 73.4 | 97.7 |
| | | | | $\text{DFR}_{\text{Tr}}^{\text{Val}}$ | 90.9 | 92.2 |
| $10^{-3}$ | $10^{-2}$ | 32 | ✓ | Base Model | 24.1 | 94.4 |
| | | | | $\text{DFR}_{\text{Tr}}^{\text{Val}}$ | 88.0 | 88.6 |
| $10^{-3}$ | $10^{-2}$ | 32 | ✗ | Base Model | 66.4 | 97.1 |
| | | | | $\text{DFR}_{\text{Tr}}^{\text{Val}}$ | 90.1 | 90.5 |
| $3\cdot10^{-3}$ | $10^{-3}$ | 32 | ✓ | Base Model | 71.5 | 98.1 |
| | | | | $\text{DFR}_{\text{Tr}}^{\text{Val}}$ | 91.9 | 93.5 |
| $3\cdot10^{-3}$ | $10^{-3}$ | 32 | ✗ | Base Model | 76.5 | 98.0 |
| | | | | $\text{DFR}_{\text{Tr}}^{\text{Val}}$ | 89.5 | 94.5 |
| $10^{-3}$ | $10^{-3}$ | 64 | ✓ | Base Model | 73.5 | 98.0 |
| | | | | $\text{DFR}_{\text{Tr}}^{\text{Val}}$ | 93.1 | 95.0 |
| $10^{-3}$ | $10^{-3}$ | 64 | ✗ | Base Model | 69.5 | 97.4 |
| | | | | $\text{DFR}_{\text{Tr}}^{\text{Val}}$ | 89.0 | 93.6 |

| Base Model Hypers | | | | Method | CelebA | |
|---|---|---|---|---|---|---|
| lr | wd | batch size | aug | | Worst(%) | Mean(%) |
| $10^{-3}$ | $10^{-4}$ | 128 | ✓ | Base Model | $46.9_{\pm2.8}$ | $95.3_{\pm0}$ |
| | | | | $\text{DFR}_{\text{Tr}}^{\text{Val}}$ | $88.3_{\pm1.1}$ | $91.3_{\pm0.3}$ |
| $10^{-3}$ | $10^{-3}$ | 128 | ✓ | Base Model | $44.3_{\pm6.4}$ | $95.2_{\pm0.1}$ |
| | | | | $\text{DFR}_{\text{Tr}}^{\text{Val}}$ | $86.2_{\pm1.2}$ | $90.8_{\pm0.7}$ |
| $10^{-3}$ | $10^{-4}$ | 128 | ✗ | Base Model | $46.7_{\pm0.0}$ | $95.3_{\pm0.1}$ |
| | | | | $\text{DFR}_{\text{Tr}}^{\text{Val}}$ | $86.9_{\pm1.1}$ | $91.6_{\pm0.2}$ |
| $10^{-3}$ | $10^{-3}$ | 128 | ✗ | Base Model | $40.6_{\pm8.7}$ | $95.1_{\pm0.2}$ |
| | | | | $\text{DFR}_{\text{Tr}}^{\text{Val}}$ | $85.6_{\pm1.4}$ | $91.8_{\pm0.5}$ |

Table 8: **Effect of base model hyper-parameters.** We report the results of $\text{DFR}_{\text{Tr}}^{\text{Val}}$ for a range of base model hyper-parameters on Waterbirds and CelebA as well as the performance of the corresponding base models. While the quality of the base model has an effect on DFR, the results are fairly robust. For a subset of configurations we report the mean and standard deviation over 3 independent runs of the base model and DFR.

| Method | Group Info | Waterbirds | | CelebA | | MultiNLI | | CivilComments | |
|---|---|---|---|---|---|---|---|---|---|
| | Train / Val | Worst(%) | Mean(%) | Worst(%) | Mean(%) | Worst(%) | Mean(%) | Worst(%) | Mean(%) |
| Base (ERM) | ✗ / ✗ | $74.9_{\pm2.4}$ | $98.1_{\pm0.1}$ | $46.9_{\pm2.8}$ | $95.3_{\pm0}$ | $65.9_{\pm0.3}$ | $82.8_{\pm0.1}$ | $55.6_{\pm0.6}$ | $92.1_{\pm0.1}$ |
| $\text{DFR}_{\text{Tr}}^{\text{Tr}}$ | ✓ / ✓ | $90.2_{\pm0.8}$ | $97.0_{\pm0.3}$ | $80.7_{\pm2.4}$ | $90.6_{\pm0.7}$ | $71.5_{\pm0.6}$ | $82.5_{\pm0.2}$ | $58.0_{\pm1.3}$ | $92.0_{\pm0.1}$ |
| $\text{DFR}_{\text{Tr}}^{\text{Val}}$ | ✗ / ✓✓ | $\mathbf{92.9}_{\pm0.2}$ | $94.2_{\pm0.4}$ | $88.3_{\pm1.1}$ | $91.3_{\pm0.3}$ | $\mathbf{74.7}_{\pm0.7}$ | $82.1_{\pm0.2}$ | $\mathbf{70.1}_{\pm0.8}$ | $87.2_{\pm0.3}$ |
| Base Model trained without minority groups | | | | | | | | | |
| Base (ERM) | ✗ / ✗ | $31.9_{\pm3.6}$ | $96.0_{\pm0.2}$ | $21.7_{\pm3.2}$ | $95.2_{\pm0.1}$ | $66.0_{\pm1.6}$ | $82.5_{\pm0.1}$ | $8.4_{\pm4.9}$ | $73.8_{\pm0.3}$ |
| $\text{DFR}_{\text{Tr-NM}}^{\text{Tr}}$ | ✓ / ✓ | $89.9_{\pm0.6}$ | $94.3_{\pm0.5}$ | $\mathbf{89.0}_{\pm1.1}$ | $91.6_{\pm0.4}$ | $73.0_{\pm0.6}$ | $82.2_{\pm0.1}$ | $66.5_{\pm0.7}$ | $85.9_{\pm0.3}$ |

Table 9: **DFR variations.** Worst-group and mean test accuracy of DFR variations on the benchmark datasets problems. $\text{DFR}_{\text{Tr}}^{\text{Val}}$ achieves the best performance across the board. Interestingly, $\text{DFR}_{\text{Tr-NM}}^{\text{Tr}}$ outperforms $\text{DFR}_{\text{Tr}}^{\text{Tr}}$ on all dataset and even outperforms $\text{DFR}_{\text{Tr}}^{\text{Val}}$ on CelebA. For all experiments, we report mean±std over 5 independent runs of the method.

convolutional layer and the fully-connected classifier layer) or the last residual block is competitive (but worse), while retraining the full network is a lot worse.

| | Number of retrains | | | | |
|---|---|---|---|---|---|
| | 1 | 3 | 5 | 10 | 20 |
| Waterbirds | $91.21_{\pm1.82}$ | $92.88_{\pm0.45}$ | $91.73_{\pm1.25}$ | $93.13_{\pm0.29}$ | $92.89_{\pm0.19}$ |
| CelebA | $85.09_{\pm1.49}$ | $88.64_{\pm1.90}$ | $87.80_{\pm1.17}$ | $88.02_{\pm1.82}$ | $88.37_{\pm2.02}$ |

Table 10: **Ablation on the number of retrains in DFR$_{\text{Tr}}^{\text{Val}}$ .** Worst group accuracy of DFR$_{\text{Tr}}^{\text{Val}}$ varying the number of logistic regression retrains on Waterbirds and CelebA: we train the logistic regression model on the validation data several times with random subsets of the data (we take all of the data from the smallest group, and subsample the other groups randomly to have the same number of datapoints) and average the weights of the learned models. Averaging more than 1 linear model leads to improved performance.

**What if we just use ImageNet features?** As a baseline, we apply DFR to features extracted by a model pretrained on ImageNet with no fine-tuning on CelebA and Waterbirds data. We report the results in Table 6 (DFR$_{\text{IN}}^{\text{Tr}}$ and DFR$_{\text{IN}}^{\text{Val}}$ lines). While on Waterbirds the performance is fairly good, on CelebA fine-tuning is needed to get reasonable performance. The results in Table 6 suggest that both ImageNet pretraining and fine-tuning on the target data are needed to train the best feature extractor for DFR.

**Robustness to base model hyper-parameters** In Table 8 we report the results of DFR$_{\text{Tr}}^{\text{Val}}$ for a range of configurations of the baseline model hyper-parameters. While the quality of the base model clearly has an effect on DFR performance, we achieve competitive results for all the hyper-parameter configurations that we consider, even when the base model performs poorly. For example, on Waterbirds with data augmentation, learning rate $10^{-3}$ and weight decay $10^{-2}$ the base model achieves worst group accuracy of $24.1\%$, but by retraining the last layer of this model with DFR$_{\text{Tr}}^{\text{Val}}$ we still achieve $88\%$ worst group accuracy.

**Ablation on the number of linear model retrains.** We study the effect of the number of logistic regression retrains on different balanced subsets of the validation set in Table 10 on Waterbirds and CelebA. We retrain the last linear layer multiple times and average the parameters of the resulting models, as described in Appendix C. In general, it is better to train more than 1 model and average their weights. This effect is especially prominent on CelebA, where a single model gets $85\%$ worst group accuracy, while the average of 3 models gets $88.6\%$. As we increase the number of linear model retrains, the improvements saturate.

**Ablation on the $\ell_1$ regularization.** We emphasize that it is beneficial to use $\ell_1$ regularization in spurious correlations benchmarks where the number of last layer features is much higher than the reweighting dataset size (e.g. in Waterbirds we use approximately 500 examples for retraining the last layer in DFR$_{\text{Tr}}^{\text{Val}}$ while the dimensionality of the penultimate layer representations in ResNet-50 is 2048). Without $\ell_1$ regularization, DFR$_{\text{Tr}}^{\text{Val}}$ achieves $87.72 \pm 0.42\%$ worst group accuracy on Waterbirds and $86.03 \pm 0.42\%$ on CelebA, as opposed to $92.9 \pm 0.2\%$ and $88.3 \pm 1.1\%$ if we choose $\ell_1$ regularization strength through cross-validation as described in Appendix C.

**Full model fine-tuning on validation.** As an additional baseline, we finetune the full model (as opposed to just the last layer) on the group-balanced validation set for 10 epochs with SGD starting from the ResNet-50 checkpoint pre-trained on ImageNet and without training on the corresponding train splits of Waterbirds and CelebA. We achieve $89.3 \pm 1.3\%$ worst group accuracy on Waterbirds and $84.4 \pm 0.5\%$ on CelebA. While these results are good relative to ERM on the standard training set, they are still significantly worse than DFR$_{\text{Tr}}^{\text{Val}}$ (see Table 2).

## C.2 Prior work assumptions

In Section 6 we compare DFR to ERM, Group DRO (Sagawa et al., 2019), JTT (Liu et al., 2021), CnC (Zhang et al., 2022), SUBG (Idrissi et al., 2021) and SSA (Nam et al., 2022). These methods differ in assumptions about the amount of group information available.

Group DRO and SUBG assume that both train and validation data have group labels, and the hyper-parameters are tuned using worst-group validation accuracy. DFR$_{\text{Tr}}^{\text{Tr}}$ and DFR$_{\text{Tr-NM}}^{\text{Tr}}$ match the setting

of these methods and use the same data and group information; in particular, these methods only use the validation set with group labels to tune the hyper-parameters.

A number of prior works (e.g. Liu et al., 2021; Creager et al., 2021; Zhang et al., 2022) sidestep the assumption of knowing the group labels on train data, but still rely on tuning hyper-parameters using worst-group accuracy on validation, and thus, having group labels on validation data. In fact, Idrissi et al. (2021) showed that ERM is a strong baseline when tuned with worst-group accuracy on validation.

Lee et al. (2022) explore the setting where they do not necessarily require group labels on validation data, but their method implicitly relies on the presence of sufficiently many minority examples in the validation set such that different prediction heads would disagree on those examples to choose the most reliable classifier.

Recent works Nam et al. (2022) and Sohoni et al. (2021) explore the setting where the group information is available on a small subset of the data (e.g. on the validation set), and the goal is to use the available data optimally, both to train the model and to tune the hyper-parameters. These methods use semi-supervised learning to extrapolate the available group labels to the rest of the training data. We consider this same setting with $\text{DFR}_{\text{Tr}}^{\text{Val}}$, where we use the validation data to retrain the last layer of the model.

## C.3   DFR VARIATIONS

Here, we discuss two additional variations of $\text{DFR}_{\hat{\mathcal{D}}}^{\hat{\mathcal{D}}}$. In $DFR_{Tr}^{Tr}$, we use a random group-balanced subset of the train data as $\hat{\mathcal{D}}$. In $DFR_{Tr\text{-}NM}^{Tr}$ (NM stands for "No Minority") we use a random group-balanced subset of the train data as $\hat{\mathcal{D}}$, but remove the minority groups ($\mathcal{G}_2, \mathcal{G}_3$ on Waterbirds and $\mathcal{G}_4$ on CelebA) from the data $\mathcal{D}$ used to train the feature extractor.

We compare different DFR versions in Table 9. All three DFR variations obtain results competitive with state-of-the-art Group DRO results on the Waterbirds data. On CelebA, $\text{DFR}_{\text{Tr}}^{\text{Val}}$ and $\text{DFR}_{\text{Tr-NM}}^{\text{Tr}}$ match Group DRO, while $\text{DFR}_{\text{Tr}}^{\text{Tr}}$ performs slightly worse, but still on par with JTT. In particular, $DFR_{Tr\text{-}NM}^{Tr}$ *matches the state-of-the-art Group DRO by using the same data to train and tune the model.* Notably, $\text{DFR}_{\text{Tr-NM}}^{\text{Tr}}$ achieves these results with the features extracted by the network trained *without seeing any examples from the minority groups*! Even without minority groups, ERM models extract the core features sufficiently well to achieve state-of-the-art results on the image classification benchmarks.

$\text{DFR}_{\text{Tr}}^{\text{Tr}}$ also significantly improves performance compared to the base model across the board, but underperforms compared to $\text{DFR}_{\text{Tr}}^{\text{Val}}$: it is crucial to retrain the last layer on new data that was not used to train the feature extractor.

On the NLP datasets, $\text{DFR}_{\text{Tr}}^{\text{Val}}$ outperforms the other variations, but in all cases all DFR variations significantly improve performance compared to the base model. Notably, on CivilComments the no minority version group of the dataset only retains the toxic comments which mention the protected identities and non-toxic comments which do not mention these identities. As a result, the base model only achieves $8.4\%$ worst group performance! $\text{DFR}_{\text{Tr-NM}}^{\text{Tr}}$ is still able to recover $66.5\%$ worst group accuracy using the features extracted by this model.

## C.4   WHY IS $\text{DFR}_{\text{TR-NM}}^{\text{TR}}$ BETTER THAN $\text{DFR}_{\text{TR}}^{\text{TR}}$?

Let us consider the second stage of $\text{DFR}_{\hat{\mathcal{D}}}^{\hat{\mathcal{D}}}$, where we fix the feature encoder $f(\cdot)$, and train a logistic regression model $\mathcal{L}$ on the dataset $f(\hat{\mathcal{D}})$, where by $f(\hat{\mathcal{D}})$ we denote the dataset with labels from the reweighting dataset $\hat{\mathcal{D}}$ and features from $\hat{\mathcal{D}}$ extracted by $f$. We then evaluate the logistic regression model $\mathcal{L}$ on the features extracted from the test data, $f(\mathcal{D}_{\text{Test}})$.

Let us use $\hat{\mathcal{M}}$ to denote a minority group in the reweighting dataset $\hat{\mathcal{D}}$, and $\mathcal{M}^{\text{Test}}$ to denote the same minority group in the test data. For $\text{DFR}_{\text{Tr-NM}}^{\text{Tr}}$ and $\text{DFR}_{\text{Tr}}^{\text{Val}}$, the model $f$ is trained without observing any data from $\hat{\mathcal{M}}$ or $\mathcal{M}^{\text{Test}}$. Assuming the datapoints in $\hat{\mathcal{M}}$ and $\mathcal{M}^{\text{Test}}$ are iid samples from the same distirbution, the distribution of features in $f(\hat{\mathcal{M}})$ and $f(\mathcal{M}^{\text{Test}})$ will also be identical.

On the other hand, with $\text{DFR}_{\text{Tr}}^{\text{Tr}}$, the minority group datapoints $\hat{\mathcal{M}}$ are used to train the feature extractor $f$. In this case, we can no longer assume that the distribution of features $f(\hat{\mathcal{M}})$ will be the

Figure 8: **DFR Variations.** Visualization of the features extracted from the reweighting dataset $\hat{D}$ and the test data for different variations of DFR on the Waterbirds data. We show projections of the 2048-dimensional features on the top-2 principal components extracted from $\hat{D}$. With $\mathrm{DFR}_{\mathrm{Tr}}^{\mathrm{Val}}$ and $\mathrm{DFR}_{\mathrm{Tr\text{-}NM}}^{\mathrm{Tr}}$, the distribution of the features for the minority groups $\mathcal{G}_2$ and $\mathcal{G}_3$ does not change between the reweighting and test data, while with $\mathrm{DFR}_{\mathrm{Tr}}^{\mathrm{Tr}}$ we see significant distribution shift.

same as the distribution of $f(\mathcal{M}^{\mathrm{Test}})$. Consequently, in $\mathrm{DFR}_{\mathrm{Tr}}^{\mathrm{Tr}}$, the logistic regression model $\mathcal{L}$ will be evaluated under *distribution shift*, which makes the problem much more challenging and leads to inferior performance of $\mathrm{DFR}_{\mathrm{Tr}}^{\mathrm{Tr}}$ on the Waterbirds data. We verify this intuition in Figure 8, where we visualize the feature embeddings for the reweighting dataset $\hat{D}$ and the test data. We see that as we predicted, the distribution of the minority group features coincides between $\hat{D}$ and test data for $\mathrm{DFR}_{\mathrm{Tr\text{-}NM}}^{\mathrm{Tr}}$ and $\mathrm{DFR}_{\mathrm{Tr}}^{\mathrm{Val}}$, while $\mathrm{DFR}_{\mathrm{Tr}}^{\mathrm{Tr}}$ shows significant distribution shift.

**What $\hat{\mathcal{D}}$ should be used in practice?**    In Table 9, the best performance is achieved by $\mathrm{DFR}_{\mathrm{Tr\text{-}NM}}^{\mathrm{Tr}}$ and $\mathrm{DFR}_{\mathrm{Tr}}^{\mathrm{Val}}$, which retrain the last layer on data that was not used in training the feature extractor. In practice, we recommend collecting a group-balanced validation set, which can be used both to tune the hyper-parameters and re-train the last layer of the model, as we do in $\mathrm{DFR}_{\mathrm{Tr}}^{\mathrm{Val}}$.

## C.5    RELATION TO TRANSFER LEARNING

Algorithmically, DFR is a special case of transfer learning, where $\mathcal{D}$ serves as the source data, and $\hat{\mathcal{D}}$ is the target data (Sharif Razavian et al., 2014). However, the motivation of DFR is different from that of standard transfer learning: in DFR we are trying to correct the behavior of a pretrained model, and reduce the effect of spurious features, while in transfer learning the goal is to learn general features that generalize well to diverse downstream tasks. For example, we can use DFR to reduce the reliance of ImageNet-trained models on the background or texture and improve their robustness to covariate shift (see Section 7), while in standard transfer learning we would typically use a pretrained model as initialization or a feature extractor to learn a good solution on a *new dataset*. In the context of spurious correlations, one would not expect DFR to work as well as it does: the only reason DFR is successful is because, contrary to conventional wisdom, *neural network classifiers are in fact learning substantial information about core features, even when they seem to rely on spurious features to make predictions*. Moreover, in Appendix Table 6 we show that transfer learning with features learned ImageNet does not work nearly as well as DFR on the spurious correlation benchmarks.

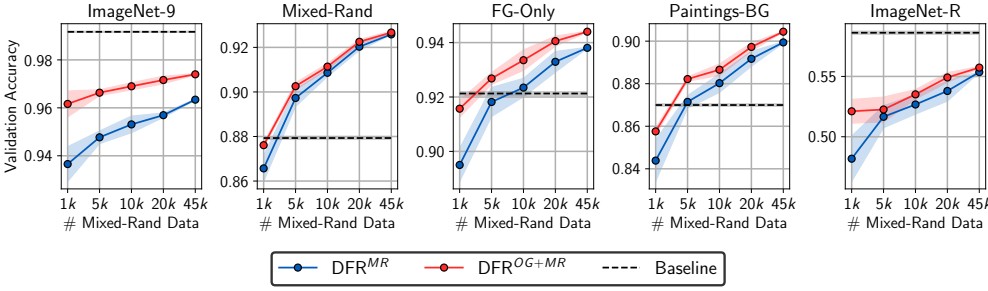

Figure 9: **ImageNet background reliance (VIT-B-16).** Performance of DFR trained on MixedRand data and MixedRand + Original data on different ImageNet-9 validation splits. All methods use an VIT-B-16 feature extractor trained on ImageNet21k and finetuned on ImageNet. DFR can reduce background reliance with a minimal drop in performance on the Original data. See Figure 3 for analogous results with a ResNet-50 feature extractor.

## D    DETAILS: IMAGENET BACKGROUND RELIANCE

Here we provide details on the experiments on background reliance in Section 7.1 and additional evaluations verifying the main results.

**Data.**    We use the ImageNet-9 dataset (Xiao et al., 2020). To test whether our models can generalize to unusual backgrounds, we additionally generate *Paintings-BG* data shown in Figure 11. For the Paintings-BG data we use the Original images and segmentation masks for ImageNet-9 data provided by Xiao et al. (2020), and combine them with random paintings from Kaggle's `painter-by-numbers` dataset available at kaggle.com/c/painter-by-numbers/ as backgrounds. For the ImageNet-R dataset (Hendrycks et al., 2021), we only use the images that fall into one of the ImageNet-9 categories, and evaluate the accuracy with respect to these categories. We show examples of images from different dataset variations in Figure 11.

**Base  model  hyper-parameters.**    We  use  a  ResNet-50  model  pretrained  on  Im-
ageNet  and  a  VIT-B-16  model  pretrained  on  ImageNet-21k  and  finetuned  on
ImageNet.      The  ResNet-50  model  is  imported  from  the  `torchvision`  pack-
age:    `torchvision.models.resnet50(pretrained=True)`.      The    VIT-B-
16  model  is  imported  from  the  `lukemelas/PyTorch-Pretrained-ViT`  pack-
age  available  at  github.com/lukemelas/PyTorch-Pretrained-ViT;  we  use  the  command:
`ViT('B_16_imagenet1k', pretrained=True)` to load the model. To extract the embed-
dings, we remove the last linear classification layer from each of the models. We preprocess the data
using the `torchvision.transforms` paclage `Compose([ Resize(resize_size),
CenterCrop(crop_size), ToTensor(), Normalize([0.485, 0.456,
0.406], [0.229, 0.224, 0.225])])`,    where    `(resize_size, crop_size)`
are equal to (256, 224) for the ResNet-50 and (384, 384) for the VIT-B-16. We do not apply any data
augmentation.

**DFR details.**    We Train DFR on random subsets of the Mixed-Rand train data of different sizes (DFR$^{MR}$) or combinations of the Mixed-Rand and Original data (DFR$^{OG+MR}$). For DFR$^{OG+MR}$  we use the same number of Original and Mixed-Rand datapoints in all experiments. As the full ImageNet-9 contains $50k$ datapoints, we train the logistic regression on GPU with a simple implementation in PyTorch (Paszke et al., 2017). We then preprocess the embeddings to have zero mean and unit standard deviation using the standard scaler `sklearn.preprocessing.StandardScaler` from the `scikit-learn` package. In each case, we compute the preprocessing statistics on the reweighting data used to train the last layer. For the experiments in this section we use $\ell_2$ regularization (as the number of datapoints is large relative to the number of observations, we do not have to use $\ell_1$). We set the regularization coefficient $\lambda$ to be 100 and train the logistic regression model with the loss $\sum_{x,y \in \mathcal{D}'} L(y, wx + b) + \frac{\lambda}{2}\|w\|^2$, where $L(\cdot, \cdot)$ is the cross-entropy loss. We use full-batch SGD with learning rate 1 and no momentum to train the model for 1000 epochs. We did not tune the $\lambda$ parameter or SGD hyper-parameters.

Figure 10: GradCAM visualizations of the features used by the baseline model and DFR$^{MR}$ on ImageNet-9.

**Results for VIT-B-16.** We report the results for the VIT-B-16 base model in Figure 9. The results are generally analogous to the results for ResNet-50: it is possible to significantly reduce the reliance of the model on the background by retraining the last layer with DFR. For the VIT, removing the background dependence hurts the performance on the Original data slightly, but greatly improves the performance on the images with unusual backgrounds (Mixed-Rand, FG-Only, Paintings-BG). Removing the background dependence does not improve the performance on ImageNet-R.

**BG modification.** For each test image we generate images with 7 modified backgrounds: MixRand (random Only-BG-T ImageNet-9 background, [1]), Paintings-BG, and constant black, white, red, green and blue backgrounds. Then, for each model from Section 7.1 we compute the percentage of the datapoints, on which changing the background with a fixed foreground does not change the predictions compared to the original image. We get $87.5\%$ for the baseline model, $92.4\%$ for DFR$^{MR}$and $93.1\%$ for DFR$^{OG+MR}$, suggesting that the DFR$^{MR}$ and DFR$^{OG+MR}$ models are significantly more robust to modifying the background.

**Prediction based on BG.** Next, for each model we evaluate the percentage of test MixRand datapoints on which the model makes predictions that match the background class and not the foreground class. We get $14.8\%$ for the baseline model, $11.2\%$ for DFR$^{MR}$ and $11.7\%$ for DFR$^{OG+MR}$. The baseline model predicts the background class significantly more frequently than the DFR models.

**GradCAM.** Finally, to gain a visual intuition into the features used by DFR$^{MR}$and the baseline models, in Figure 10 we make GradCAM feature visualizations on three images from ImageNet-9. We use the `pytorch-grad-cam` package (Gildenblat & contributors, 2021) available at github.com/jacobgil/pytorch-grad-cam. While the baseline model uses the background context, the DFR$^{MR}$ features are more compact and focus on the target object, again suggesting that DFR reduces the reliance on the background information.

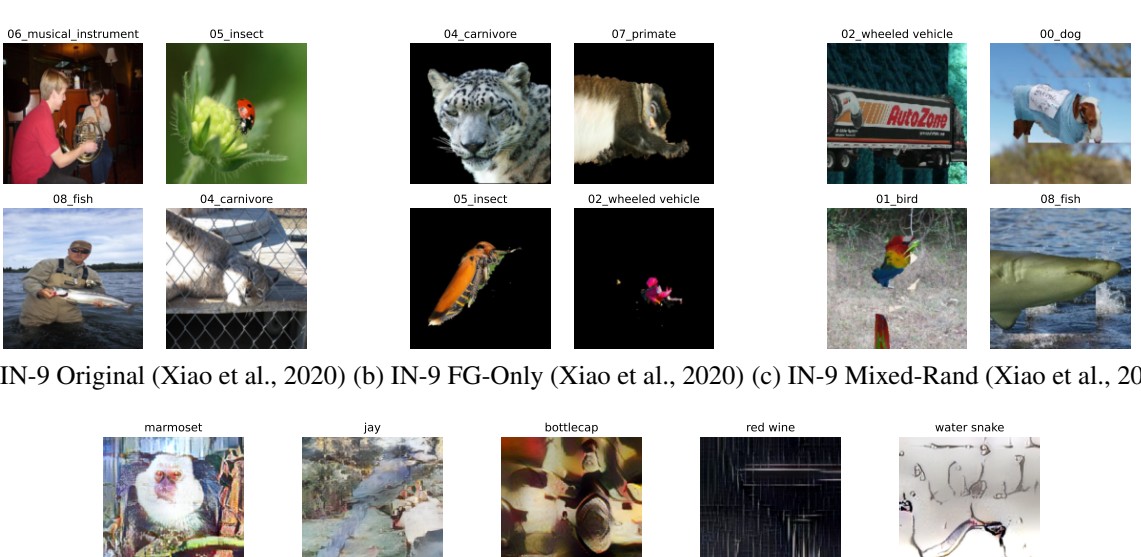

(a) IN-9 Original (Xiao et al., 2020) (b) IN-9 FG-Only (Xiao et al., 2020) (c) IN-9 Mixed-Rand (Xiao et al., 2020)

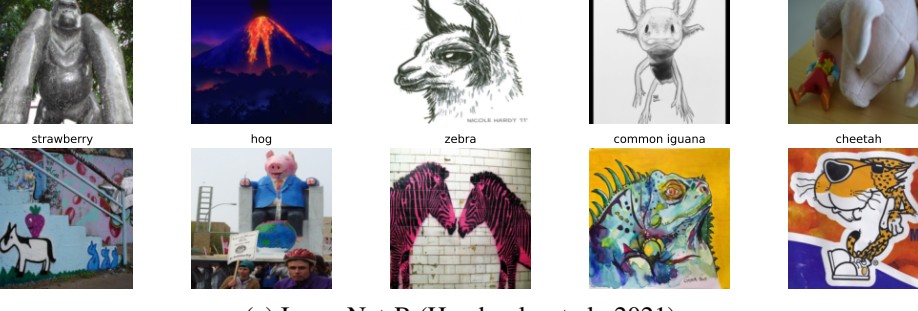

(d) Stylized ImageNet (Geirhos et al., 2018)

(e) ImageNet-R (Hendrycks et al., 2021)

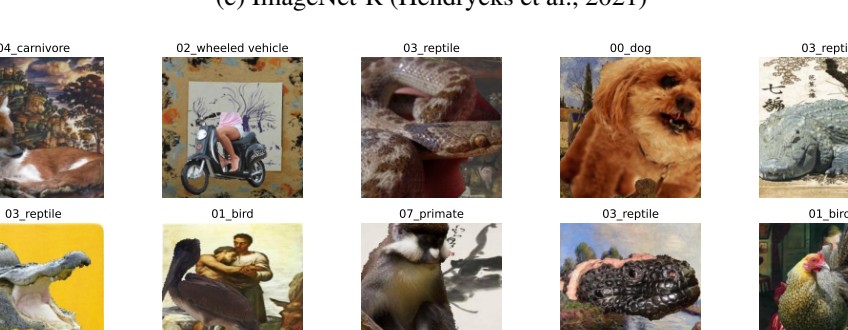

(f) ImageNet-9, Paintings-BG (ours)

Figure 11: **ImageNet variations.** Examples of datapoints from the ImageNet variations used in the experiments.

# E  DETAILS: IMAGENET TEXTURE BIAS DETAILS

Here we provide details on the experiments on texture bias in Section 7.

**Data.** We generate the stylized ImageNet (SIN) data following the instructions at github.com/rgeirhos/Stylized-ImageNet (Geirhos et al., 2018). For evaluation, we use ImageNet-C (Hendrycks & Dietterich, 2019) and ImageNet-R datasets (Hendrycks et al., 2021). For ImageNet-C we report the average performance across all 19 corruption types and 5 corruption intensities. We show examples of stylized ImageNet images in Figure 11.

**Base model hyper-parameters.** We use the same ResNet-50 and VIT-B-16 models as described in Appendix D.

**DFR details.** We train DFR using the embeddings of the original ImageNet (IN), styl­ized ImageNet (SIN) and their combination (IN+SIN) as the reweighting dataset. We pre­process the base model embeddings by manually subtracting the mean and dividing by standard deviation computed on the reweighting data used to train DFR; we did not use `sklearn.preprocessing.StandardScaler` due to the large size of the datasets ($1.2M$ datapoints for IN and SIN; $2.4M$ datapoints for IN+SIN). We train the logistic regression for the last layer on a single GPU with a simple implementation in PyTorch. We use SGD with learning rate 1, no momentum, no regularization and batch size of $10^4$ to train the model for 100 epochs. We tuned the batch size (in the range $\{10^3, 10^4, 10^5\}$) and picked the batch size that leads to the best performance on the ImageNet validation set ($10^4$). We did not tune the other hyper-parameters.

**Results for VIT-B-16.** We report the results for the VIT-B-16 base model in Table 11. For this model, baselines trained from scratch on IN+SIN and SIN are not available so we only report the results for the standard model trained on ImageNet21k and finetuned on ImageNet; for DFR we report the results using IN+SIN as the reweighting dataset. Despite the large-scale pretraining on ImageNet21k, we find that we can still improve the shape bias ($36\% \rightarrow 39.9\%$) as well as robustness to ImageNet-C corruptions ($49.7\% \rightarrow 52\%$ Top-1 accuracy). On the original ImageNet and ImageNet-R the performance of DFR is similar to that of the baseline model.

**Detailed texture bias evaluation.** To provide further insight into the texture bias of the models trained with DFR, in Figure 12 we report the fraction of shape and texture decisions for different classes following Geirhos et al. (2018). We produce the figure using the `modelvshuman` codebase available at github.com/bethgelab/model-vs-human (Geirhos et al., 2021). We report the results for both DFR models and models trained from scratch on IN, SIN and IN+SIN as well as the ShapeResNet-50 model and humans (results from Geirhos et al. (2018)). When trained on the same data, models trained from scratch achieve a higher shape bias than DFR models, but DFR can still significantly improve the shape bias compared to the base model trained on IN.

**Detailed ImageNet-C results.** In Figure 13, we report the Top-1 accuracy for DFR models and models trained from scratch on IN, SIN and IN+SIN on individual ImageNet-C datasets. We use the ResNet-50 base model. The model trained from scratch on IN+SIN provides the best robustness across the board, but DFR trained on IN+SIN also provides an improvement over the baseline RN50(IN) model on many corruptions.

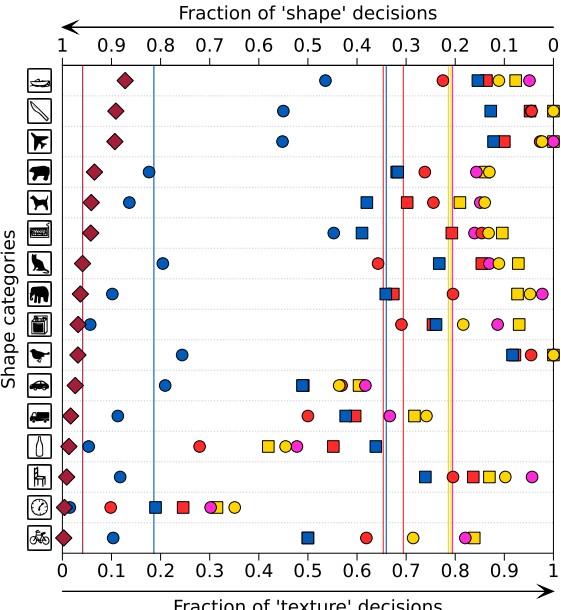

Figure 12: **Shape-texture bias report.** Detailed report of the shape-texture bias generated using the `model-vs-human` codebase (`https://github.com/bethgelab/model-vs-human`) (Geirhos et al., 2021). The plot shows the fraction of the decisions made based on shape and texture information respectively on examples with conflicting cues (Geirhos et al., 2018). The brown diamonds (◇) show human predictions and the circles (○) show the performance of ResNet-50 models trained on different datasets: ImageNet (IN, yellow), Stylized ImageNet (SIN, blue), ImageNet + Stylized ImageNet (IN+SIN, red), ImageNet + Stylized ImageNet Finetuned on ImageNet (IN+SIN→IN, pink); these models are provided in the `model-vs-human` codebase. For each dataset (except for IN+SIN→IN) we report the results for DFR using an ImageNet-trained ResNet-50 model as a feature extractor with squares □ of the corresponding colors. Reweighting the features in a pretrained model with DFR we can significantly increase the shape bias: DFR trained on SIN (blue squares) virtually matches the shape bias of the model trained from scratch on IN+SIN (red circles). However, the model trained just on SIN (blue circles) from scratch still provides a significantly higher shape bias, that we cannot match with DFR.

| Method | Training Data | Shape bias (%) | Top-1 Acc (%) / Top-5 Acc (%) | | |
|---|---|---|---|---|---|
| | | | ImageNet | ImageNet-R | ImageNet-C |
| VIT-B-16 | IN21k + IN | 36 | 79.2/95.0 | 29.1/42.0 | 49.7/69.3 |
| DFR | IN+SIN | 39.9 | 79.7/94.5 | 29.0/41.4 | 52.0/71.0 |

Table 11: **Texture-vs-shape bias results for VIT-B-16.** Shape bias, top-1 and top-5 accuracy on ImageNet validation set variations for VIT-B-16 pretrained on ImageNet21k and finetuned on ImageNet and DFR using this model as a feature extractor and reweighting the features on combined ImageNet and Stylized ImageNet datasets. By retraining just the last layer with DFR, we can increase the shape bias compared to the feature extractor model and improve robustness to covariate shift on ImageNet-C.

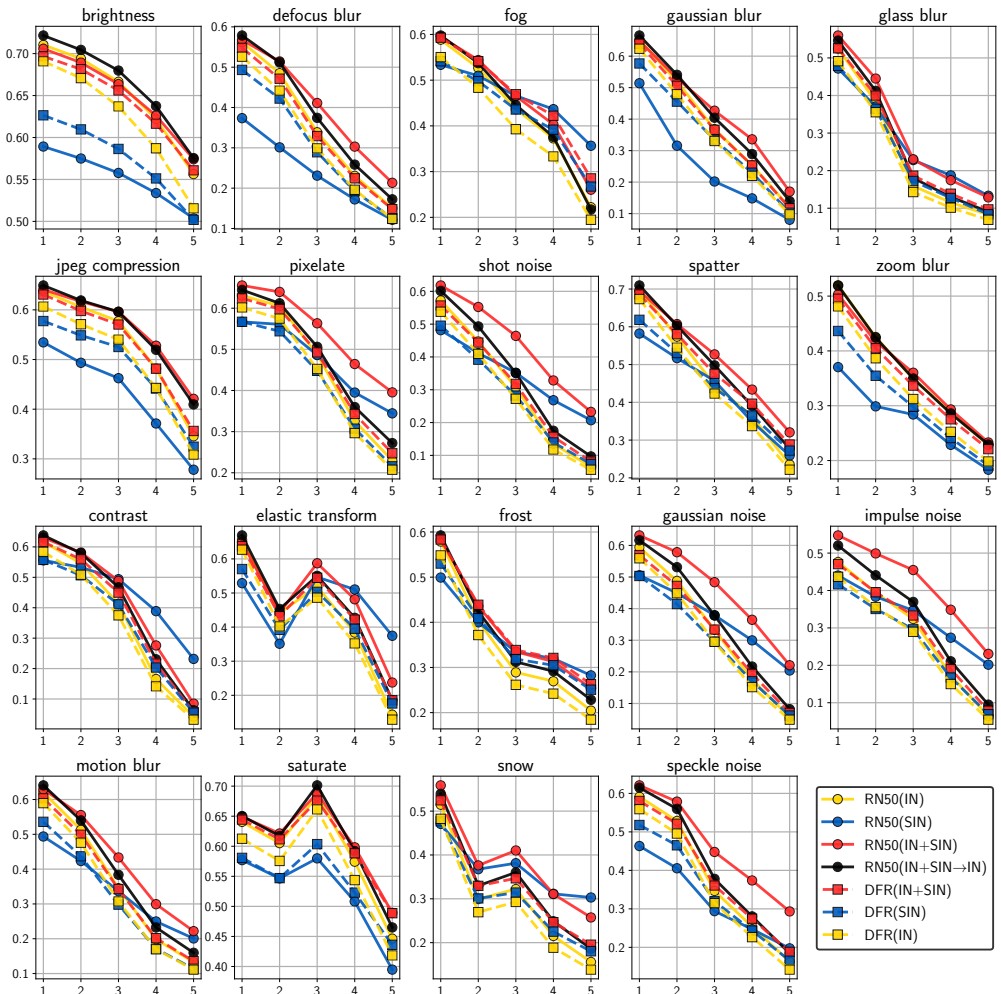

Figure 13: **ImageNet-C results.** Results of ResNet-50 models trained on different ImageNet variations (shown in circles) and DFR using an ImageNet-trained ResNet-50 model as a feature extractor on ImageNet-C and ImageNet-R datasets. Each panel corresponds to a different corruption, and the horizontal axis represents the corruption intensity. Retraining the last layer on ImageNet-Stylized (DFR(SIN), red squares) improves robustness to ImageNet-C corruptions compared to the base model (RN50(IN), blue circles), but does not match the robustness of a model trained from scratch on SIN or IN+SIN.

# F FURTHER DISCUSSION

**Spurious correlations and representation learning.** Prior work has often associated poor robustness to spurious correlations with the quality of *representations* learned by the model (Arjovsky et al., 2019; Bahng et al., 2020; Ruan et al., 2021) and suggested that the entire model needs to be carefully trained to avoid relying on spurious features (e.g. Sagawa et al., 2019; Idrissi et al., 2021; Liu et al., 2021; Zhang et al., 2022; Sohoni et al., 2021; Pezeshki et al., 2021; Lee et al., 2022). Our work presents a different view: *representations learned with standard ERM even without seeing any minority group examples are sufficient to achieve state-of-the-art performance on popular spurious correlation benchmarks.* The issue of spurious correlations is not in the features extracted by the models (though the representations learned by ERM can be improved (e.g., Hermann & Lampinen, 2020; Pezeshki et al., 2021)), but in the weights assigned to these features. Thus we can simply re-weight these features for substantially improved robustness.

**Practical advantages of DFR.** DFR is extremely simple, cheap and effective. In particular, DFR has only one tunable hyper-parameter — the strength of regularization of the logistic regression. Furthermore, DFR is highly robust to the choice of base model, as we demonstrate in Appendix Table 8, and does not require early stopping or other highly problem-specific tuning such as in Idrissi et al. (2021) and other prior works. Moreover, as DFR only requires re-training the last linear layer of the model, it is also extremely fast and easy to run. For example, we can train DFR on the $1.2M$-datapoint Stylized ImageNet dataset in Section 7 on a single GPU in a matter of minutes, after extracting the embeddings on all of the datapoints, which only needs to be done once. On the other hand, existing methods such as Group DRO (Sagawa et al., 2019) require training the model from scratch multiple times to select the best hyper-parameters, which may be impractical for large-scale problems.

**On the need for reweighting data in DFR.** Virtually all methods in the spurious correlation literature, even the ones that do not explicitly use group information to train the model, use group-labeled data to tune the hyper-parameters (we discuss the assumptions of different methods in prior work in Appendix C.2). If a practitioner has access to group-labeled data, we believe that they should leverage this data to find a better model instead of just tuning the hyper-parameters. Finally, we note that DFR can be easily combined with methods that automatically estimate group labels (e.g. Liu et al., 2021; Creager et al., 2021; Sohoni et al., 2021): we can retrain the last layer using these estimated labels instead of ground truth.

**Future work.** There are many exciting directions for future research. DFR largely reduces the issue of spurious correlations to a linear problem: how do we train an optimal linear classifier on given features to avoid spurious correlations? In particular, we can try to avoid the need for a balanced reweighting dataset by carefully studying this linear problem and only using the features that are robustly predictive across all of the training data. We can also consider other types of supervision, such as saliency maps (Singla & Feizi, 2021) or segmentation masks to tell the last layer of the model what to pay attention to in the data. Finally, we can leverage better representation learning methods (Pezeshki et al., 2021), including self-supervised learning methods (e.g. Chen et al., 2020; He et al., 2021), to further improve the performance of DFR.

# G COMPARISON TO KANG ET AL. (2019)

We compare $\mathrm{DFR}_{\mathrm{Tr}}^{\mathrm{Val}}$ and $\mathrm{DFR}_{\mathrm{Tr}}^{\mathrm{Tr}}$ to LWS and cRT methods proposed in Kang et al. (2019), and perform other related ablations in Table 12. We discuss the LWS and cRT methods in detail as well as their algorithmic differences with DFR in Section 3 and Appendix A. We adapt the original implementation for LWS and cRT[4].

In addition to LWS and cRT, we evaluate last layer re-training on the training and validation data with group-balanced data sampling. These methods serve as intermediate variations between DFR and cRT, as they use balanced sampling instead of subsampling compared to DFR, but use group information instead of class information compared to cRT. For DFR, we report the performance of $\mathrm{DFR}_{\mathrm{Tr}}^{\mathrm{Tr}}$ and $\mathrm{DFR}_{\mathrm{Tr}}^{\mathrm{Val}}$ .

We train LWS, cRT and last layer re-training variations for 10 epochs with SGD, using cosine learning rate schedule decaying from $0.2$ to $0$; DFR implementation details can be found in Appendix C.

---

[4]`https://github.com/facebookresearch/classifier-balancing`

| Method | Class or group balancing | Data | Balancing | Waterbirds WGA | CelebA WGA |
|--------|--------------------------|------|-----------|----------------|------------|
| LWS (Kang et al., 2019) | Class | Train | Balanced sampling | $40.03_{\pm 8.07}$ | $35.55_{\pm 14.4}$ |
| cRT (Kang et al., 2019) | Class | Train | Balanced sampling | $74.48_{\pm 1.5}$ | $52.88_{\pm 5.96}$ |
| Re-training last layer | Group | Train | Balanced sampling | $76.48_{\pm 1.24}$ | $56.11_{\pm 4.77}$ |
| Re-training last layer | Group | Validation | Balanced sampling | $89.21_{\pm 1.04}$ | $67.66_{\pm 2.14}$ |
| $\text{DFR}_{\text{Tr}}^{\text{Tr}}$ | Group | Train | Subsampling | $90.2_{\pm 0.8}$ | $80.7_{\pm 2.4}$ |
| $\text{DFR}_{\text{Tr}}^{\text{Val}}$ | Group | Validation | Subsampling | $\mathbf{92.9_{\pm 0.2}}$ | $\mathbf{88.3_{\pm 1.1}}$ |

Table 12: **Comparison to LWS and cRT from Kang et al. (2019) and ablations.** We compare $\text{DFR}_{\text{Tr}}^{\text{Val}}$ and $\text{DFR}_{\text{Tr}}^{\text{Tr}}$ to LWS and cRT proposed in Kang et al. (2019) for long-tail classification, as well as variations of last layer retraining on Waterbirds and CelebA dataset. The methods differ in (1) the data (train or validation) used for retraining last layer or logits scales in LWS, (2) whether class or group labels are used for balancing the dataset, and (3) the type of balancing which is either subsampling the dataset or using a balanced dataloader which first selects the class or group label uniformly at random and then samples an example from that class or group. Notice the significant improvement that we gain in terms of WGA as we change the data from train to validation, and as we change the balancing from balanced sampling to subsampling. $\text{DFR}_{\text{Tr}}^{\text{Val}}$ performs best compared to other methods.

Since the original LWS and cRT methods were proposed to address the class imbalance problem, they perform poorly in terms of the worst group accuracy in the spurious correlation setting. LWS performs especially poorly, as it only retrains a single scaling parameter per class, which is not sufficient to remove the reliance on spurious features. Re-training the last layer with balanced data sampling with respect to the group labels does improve performance compared to these original methods from Kang et al. (2019) as well as ERM, but underperforms compared to both DFR versions. This ablation highlights the importance of subsampling compared to balanced sampling (see also Idrissi et al., 2021)[5].

$\text{DFR}_{\text{Tr}}^{\text{Val}}$ achieves the best performance across the board. To sum up, for optimal performance, it is important to use held-out data (as opposed to re-using the train data) and to perform data subsampling according to group labels (as opposed to using group-balanced data sampling).

---

[5]The difference is less pronounced on Waterbirds with retraining on the validation set because the validation set is relatively group-balanced. In CelebA the group distribution is the same in validation and training sets, so subsampling performs much better than group-balanced sampling.

