# OpenReview forum: "Last Layer Re-Training is Sufficient for Robustness to Spurious Correlations"
_ICLR.cc/2023/Conference — ICLR 2023 notable top 25%_

### Official Review · Reviewer_AQfF · 2022-10-24

**Confidence:** 3
**Correctness:** 4
**Technical Novelty And Significance:** 3
**Empirical Novelty And Significance:** 3
**Recommendation:** 6

**Clarity, Quality, Novelty And Reproducibility:**

The clarity and representation of the paper are good. The study is supported by detailed evaluations and also analysis. The findings are also novel according to me.

**Strength And Weaknesses:**

Strength:

1. The paper is well-written and easy to follow.
2. The proposed approach is simple to implement yet effective in combating spurious correlations.
3. Experimental analysis in Section 4 is interesting and provides new findings.
4. Empirically achieves SOTA performance.

Weaknesses:
1. Requires access to group-labeled samples. Although Table 5 shows it is a common assumption in the literature.

Q1. It is difficult to understand why using group-labeled samples from training data performs worse than using group-labeled validation data. Are the number of samples in different groups used for reweighting kept similar for both DFR_Tr and DFR_val ?



**Summary Of The Paper:**

The paper studies the problem of spurious correlations. Authors propose to improve robustness against spurious correlations through a simple approach (Deep Feature Reweighting): specifically re-train the last layer of a neural network using a small set of reweighting
data where the spurious correlation does not hold. Further, they also show that despite relying on spurious correlations, the model still learns the core/invariant features. Empirically their proposed method achieves state-of-the-art performance.

**Summary Of The Review:**

The paper is well-organized, well-motivated, and supported by empirical evaluations. Hence, I recommend acceptance.

---

> ### Author Response · Authors · 2022-11-17
> **Author Response to Reviewer AQfF**
>
> Thank you for your supportive feedback!
>
> ## DFR on group-balanced train vs validation
>
> Thank you for your question, we provide a detailed analysis and explanation in Appendix C.3, C.4 and in Figure 8. Intuitively, we can explain this observation as follows.
>
> The feature extractor is trained to minimize the loss on the training data. Consequently, the distribution of the features (inputs to the last layer) extracted by the model is different on the training data compared to test. This behavior is analogous to the train accuracy being higher than test accuracy: in both cases the reason is that the model overfits to the training data.
> On the other hand, the distribution of the features on the validation data is the same as on test (analogously, validation accuracy matches test accuracy).
>
> As a result, when we retrain the last layer on the training data features, we experience a distribution shift when we apply the last layer to the test data features. On the other hand, the validation and test data features are iid.
>
> In Figure 8, we show a visualization of the features extracted by the ERM-trained model on Waterbirds, and confirm the intuition above.
>
> ## Summary
>
> Please let us know if you have any further questions! See also our general response, where we describe additional experiments and updates to the paper. We hope that you will consider increasing your score in light of our response.

---

### Official Review · Reviewer_8GtD · 2022-10-25

**Confidence:** 4
**Correctness:** 3
**Technical Novelty And Significance:** 3
**Empirical Novelty And Significance:** 3
**Recommendation:** 6

**Clarity, Quality, Novelty And Reproducibility:**

I have already commented on the novelty of the work in the previous section. In a nutshell, there is significant overlap between this work and recent work by Rosenfeld et al. (2022), as well as with, to a lesser extent,  Kang et al. (2019). While to me the similarity does not invalidate this work, I would argue that a more detailed discussion of overlap would make the paper stronger and more comprehensible within the related literature.

Regarding clarity, the majority of the paper is clearly written and easy to follow. However, I noticed a decline in the clarity of the text in Section 7, where the paragraphs become denser and the results harder to interpret. For instance, Table 2 is hard to read due to the large amount of numbers in the table. Furthermore, although I did not carefully reviewed the whole appendix, I also found it substantially less clear than the main paper. Finally, I would like to mention that the mixture of different data sets and methods in different sections of the paper also impacts the clarity of the paper. In order to help the reader navigate the paper, I would consider including a summary figure of table of the experimental setup.

The appendix contains details about the experimental setup, in terms of reproducibility, but I am not aware of any mention of the availability of code and data to reproduce the results.

I judge the quality of the paper as sufficient, in terms of the methods chosen and designed to evaluate the hypothesis, and in my opinion the conclusions are supported by the results.

**Details Of Ethics Concerns:**

The paper makes use of data sets with human faces.

**Strength And Weaknesses:**

### Strengths

This paper is conceptually simple, therefore easy to follow, and it is also mostly well-written. As an important strength, I highlight that the paper provides convincing empirical evidence that neural networks trained with empirical risk minimisation (ERM) are able to learn feature extractors of complex features, even in the presence of strong correlation of the labels with spurious features and the weights of the output layer mostly depend on the latter. In particular, the evidence come from two results: one, that such models obtain good (but far from optimal) performance on test sets without spurious correlations; two, that reweighting the output layer by training on a balanced data set achieves near optimal results. I think this strong evidence sharpens our understanding on the representation learning of neural networks trained with ERM and may inspire interesting avenues for future improvements on the generalisation of deep learning methods.

### Weaknesses

In view of what I consider the main strength of the paper, it is also fair to highlight that this result is not completely novel, since it has been already suggested by Rosenfeld et al. (2022), a paper that the authors cite and briefly summarise. I belief that the overlap is substantial enough to deserve a more detailed discussion of the similarities and differences between the two works.

Besides this comment on the novelty, in my opinion the paper places much importance on the proposed method to retrain the last layer of a pre-trained model by using a relatively small but controlled and balance data set. While the method proves to be effective at reducing the reliance on spurious features at the classification layer, the wide applicability of the method (and other related methods) is questionable, in that it requires both prior identification of the spurious features, as well as access to a balanced data set for re-training. I would argue that in general the main problem with spurious correlations is that they may be very hard to spot or identify a priori.

Related to the previous concern, in Section 4.2, the authors propose to use the "decoded accuracy" (accuracy after re-training the classification on a group-balanced validation set) as a way to analyse the reliance on "high-quality representations" (as opposed to the accuracy on the core-only set). However, how can the authors guarantee that the balanced validation set does not contain other spurious correlations (as in the example of the zero-vs-ones provided in the paragraph)?

**Summary Of The Paper:**

This paper tackles the concern that neural networks trained for classification exhibit a bias towards learning spurious correlations that can negatively impact generalisation. In particular, this paper empirically demonstrates that despite learning spurious features and assigning them larger weights, neural networks can also learn more complex features simultaneously. Based on this observation, the authors propose a method, named Deep Feature Reweighting (DFR), to retrain the classification layer of a deep neural network with a small, controlled, balanced data set to reweight the importance of spurious and complex features.

**Summary Of The Review:**

My overall impression of this paper is positive, since I believe the results provide strong evidence of the capability of neural networks to learn complex representations, even when the signal of spurious features is highly correlated with the labels. While this has been shown before and future work should shed more lights on the conditions when this is the case, among other open questions, I believe that this paper advances our understanding of the representations learnt by neural networks.

---

> ### Author Response · Authors · 2022-11-17
> **Author Response to Reviewer 8GtD (Part 1/2)**
>
> Thank you for your supportive feedback!
> Based on your feedback we have updated our paper and **added a more detailed discussion of the concurrent work of  Rosenfeld et al and the differences with our work. Moreover, we have also added a detailed discussion on Kang et al and additional ablation studies**.
>
> ## Relation to Rosenfeld et al. and Kang et al.
>
> First, we note that our work was completed *concurrently*  and independently from Rosenfeld et al. The first version of our paper was submitted to another conference *within two weeks* after Rosenfeld et al appeared on arXiv. Neither our work nor Rosenfeld et al have been published yet. Consequently, our work and Rosenfeld et al should be viewed as concurrent, and should not weigh in the evaluation of our paper. However, we are still happy to provide a more extensive discussion.
>
> While our works make similar high-level observations, they are actually complementary to each other. In particular, our works don’t share any datasets, experiments or problem settings.
>
> Rosenfeld et al. focus on domain generalization, where the goal is to train a model that generalizes to unseen domains. In this setting, we know the domain labels on the train, and we have no data from the test domains. In spurious correlations, the goal is to train a model that does not rely on spurious features. We typically have examples from all the test groups in train, but the groups are highly imbalanced.
>
> As in domain generalization we do not have access to target domain data, Rosenfeld et al refer to last layer retraining on the target domain as “cheating” (see e.g. the caption of Figure 1 in their paper). Consequently, they propose a different method, DARE, which is very different from DFR. DARE estimates means and covariances for each domain to whiten out the features. On test, they apply approximate whitening to a new domain. They don’t use test domain data to retrain the last layer. On the other hand, DFR uses an ERM-trained feature extractor and simply retrains the last layer on a group-balanced reweighting dataset.
>
> To sum up, the observations from our work and concurrent work by Rosenfeld et al are complimentary. We both show that ERM learns the core features well, but the settings, methods and experiments are different. In particular, it’s not trivial to apply DARE to spurious correlations or DFR to domain generalization.
>
> **Based on your feedback, we have added a detailed discussion of Rosenfeld et al to the Related Work section.**
>
> We have also added a discussion of the work by Kang et al, that you mention. This work considers the class imbalance (long tail classification) setting, and the method that they propose is also significantly different from DFR: (1) they retrain the last layer on the training data while we use a validation set, (2) they use class-balances sampling, while we subsample the dataset (based on groups) and then use uniform sampling. We provide a detailed discussion of these and other differences in [our response to reviewer xHWG](https://openreview.net/forum?id=Zb6c8A-Fghk&noteId=tEWixwfHIq), with several new ablations.
>
>
> ## The need for reweighting data
>
> As you correctly pointed out, prior works also rely on the assumption of (1) knowing a priori what exactly the spurious feature is, and (2) having group labels on training data (at least a subset of train) and validation data. Even the works which do not use group labels on train, use worst group accuracy on validation to optimize hyperparameters [e.g. JTT, Liu et al.].
>
> In order to make sure we are robust to biases and spurious correlations, we have to know what they are, in general. Indeed, without any prior information on the core vs spurious features, the problem of disambiguating core against spurious features may be underspecified in the cases of very high spurious correlations. In the extreme case when the spurious correlation strength is 100%, it is impossible to disambiguate between core and spurious features without additional assumptions.
>
> The reweighting dataset is group-balanced with respect to a particular spurious feature so in theory it may contain other spurious correlations. The same consideration applies to virtually any other method in group robustness setting, since group-labeled train or validation sets may contain other spurious correlations that are not captured by the group labels. We discuss the assumptions on the group information in many popular group robustness methods in Appendix C.2.
>
> ## Source code
>
> We have attached the source code for experiments to the supplementary material. All datasets used in this paper are publicly available, and we also provide instructions on how to access them.

---

> > ### Author Response · Authors · 2022-11-17
> > **Author Response to Reviewer 8GtD (Part 2/2)**
> >
> > ## Summary
> >
> > We agree with you that our paper advances the understanding of representation learning in neural networks. We hope we addressed your concerns regarding the novelty of the work. Also based on your feedback, we removed $\text{DFR}_{\text{Tr}}^{\text{Tr}}$ from Table 2 and simplified Table 3 significantly to make the results easier to read. If you have any further questions, please don’t hesitate to ask.
> >
> > We have also summarized our contributions and updates in our general post. We have put a significant effort into addressing your questions and updating the paper, and really hope you will recognize the significance and timeliness of this work.
> >
> > We hope that you will consider raising your score in light of our response.
> >
> >  **References**
> >
> > [Rosenfeld et al. 2022] [*Domain-Adjusted Regression or: ERM May Already Learn Features Sufficient for Out-of-Distribution Generalization*](https://arxiv.org/abs/2202.06856)
> >
> > [Kang et al., 2019]
> > [*Decoupling Representation and Classifier for Long-Tailed Recognition*](https://arxiv.org/abs/1910.09217)
> >
> > [Liu et al., 2021] [*Just Train Twice: Improving Group Robustness without Training Group Information*](https://arxiv.org/abs/2107.09044)

---

### Official Review · Reviewer_v7pb · 2022-10-25

**Confidence:** 4
**Correctness:** 3
**Technical Novelty And Significance:** 2
**Empirical Novelty And Significance:** 3
**Recommendation:** 6

**Clarity, Quality, Novelty And Reproducibility:**

The paper is well written. Most sections are easy to follow and understand.  The experiments are well design and provide sufficient details on the setup for reproducibility.

**Strength And Weaknesses:**

Strengths
- The proposed idea of retraining the linear head is simple and natural
- The paper conducts extensive well-designed studies and the method demonstrates very competitive performance compared to recent works on worst group robustness.

Weaknesses
- The phenomenon that pre-trained networks still learn causal features even under severe spurious correlation in the training set is surprising. No theoretical explanations are provided to explain the phenomenon.

**Summary Of The Paper:**

The paper proposes a surprisingly simple approach for mitigating spurious correlation via linear probing on a balanced dataset. The paper conducts comprehensive empirical studies on worst group robustness benchmarks and ImageNet variants.

**Summary Of The Review:**

The paper provides a simple method for mitigating spurious correlation. Although no theoretical analysis is provided, the paper conducts comprehensive empirical studies and is well written.

---

> ### Author Response · Authors · 2022-11-17
> **Author Response to Reviewer v7pb**
>
> Thank you for your supportive feedback!
>
> Our work provides a *comprehensive empirical study*. We show that the phenomenon holds very generally, including results in the shortcut learning scenarios, on a variety of spurious correlation benchmarks in both vision and NLP, and even in cases when no minority group examples are present in the training data (Appendix C.2).
>
> We do not think the fact that some of our observations are surprising should be held against the paper: we provide a simple and practical baseline that performs on par with sophisticated state-of-the-art methods for spurious correlations. The fact that the results are surprising means that our contribution is non-trivial, and we learned important insights.
>
> We also provide many experiments aimed at *understanding* the phenomena that we observe. In Appendix B.4 we describe logit additivity, which provides further insights into the results of Section 4.1. In Appendix C.4 and Figure 8 we explain why retraining the last layer on held-out data performs much better than retraining it on the training data.
>
> While we do not provide a formal theoretical explanation of the fact that models learn high-quality representations of the core features, we can provide an intuitive explanation. The spurious features are sufficient to classify the majority of the data, but not the minority groups, so the model has to still learn the core features which are needed to classify minority examples. Moreover, in some cases, the spurious features can be ambiguous: for example, we could find Waterbirds datapoints with ambiguous backgrounds. For these datapoints, similarly to the minority datapoints, the model is incentivised to learn the core features.
>
> The last layer retraining is necessary because the model by default finds a good classifier on the training group distribution. With standard ERM training, the model is incentivised to make accurate predictions on the majority groups more than on the minority groups. With last layer retraining, we assign equal importance to the accuracy on the minority and majority examples. We leave a formal mathematical treatment of this intuition as an exciting direction for future work.
>
> Please feel welcome to ask any further questions. We review our contributions, which we believe are significant and timely, as well as updates to the paper in the general response to all reviewers. We hope you will consider raising your score in light of our response!

---

### Official Review · Reviewer_xHWG · 2022-10-27

**Confidence:** 4
**Correctness:** 3
**Technical Novelty And Significance:** 2
**Empirical Novelty And Significance:** 3
**Recommendation:** 8

**Clarity, Quality, Novelty And Reproducibility:**

The paper is well written. All the details seem to be present for reproducibility although many are deferred to the appendix. As mentioned above, some statements might need to be rephrased to represent the technique and the results more candidly. The novelty seems limited to the observation rather than a new algorithm. That is still a valuable contribution but it needs to be presented accordingly.

**Strength And Weaknesses:**

Strength
- The paper is well written and tackles an important problem: the effect of spurious correlation on ML models.
- The analysis is interesting.
- The algorithm is simple yet effective.

Weaknesses
- The biggest weakness in my opinion is the similarity with the idea presented in “DECOUPLING REPRESENTATION AND CLASSIFIER FOR LONG-TAILED RECOGNITION” [Kang2019], concretely the algorithm that Kang et al. refer to as Learnable Weight Scaling (LWS). The authors of the current paper did cite this work and state that the two works focus on different settings. While this might be true (as LWS focuses on long tail while DFR focuses on spurious correlation) in practice the algorithms seem identical. If I am correct, then the novelty of the current paper is mostly in the observations and analysis rather than in proposing a “new” algorithm. This should be stated and in such case there would be no need to introduce a new “name” for an existing algorithm. If the algorithms are similar, but not identical, then the difference with LWS should be described in more detail and the empirical comparison should appear in Table 2.
- Ablation studies are missing and they could make the paper stronger and the finding more insightful, especially given the similarity with [Kang2019] additional ablation studies would increase the contribution. See below for some suggestions.
- Some statements might need to be rephrased to represent the technique and the results more candidly. Se below for some suggestions.

Details
1. “In order to make use of more of the available data, we train logistic regression 10 times using different random balanced subsets of the data, and average the weights of the learned models”. If averaging the model leads to important gain and it is what it is mostly reported in the experiments than it should described in Figure 1 and in the intro. This is actually a missing ablation study to understand the contribution of the weighting vs the simple rescaling. How much of the gain is related to the averaging of multiple models and how much is the use of a balanced unseen dataset?
2. The authors say “we initialize the model with weights pretrained on ImageNet”. This naturally makes one wonder if the presence of core features is “just” due to this pre-training or if a model will learn core features even without ImageNet pre-training. It seems that this pre-training affects experiments in section 4.1 but possibly not 4.2? If this is the case I’d invite the author to state this important difference in the main paper (rather than differ this detail to the appendix). I’d be curious to see the results of 4.1 without pre-training, if confirmed I think the message would be stronger.
3. The conclusions from 4.1 are too general given the pre-training step “we conclude that while the models trained on the Original data make use of background information to make predictions, they still learn the features relevant to classifying the birds almost as well as the models trained on the data without spurious correlations”. Without proving the point above (what happens when one trains only on the dataset affected by spurious correlation - without pre-training) this should be re-stated as “we conclude that while models that are pre-trained on ImageNet and fine-tuned on the Original data make use of background information to make predictions, they still learn the features relevant to classifying the birds almost as well as the models trained on the data without spurious correlations”
4. Related to the above point. When using a model pre-trained on imagenet, what happens if one only fine-tunes directly on the validation set (skipping the training on the larger but biased set)?
5. “For all DFR variations, the size of the reweighting set Dˆ is small relative to the number of features”  could be stated more precisely: what does small mean in this context?
6. I suggest to de-emphisize if possible  DFR^train_train. It is a good experiment to show the need for the use of a new set of data. However, It is a bit distracting when read in the current order. I’d consider introducing it after the main result, as an ablation study on the use of the new vs old data. Currently it is presented at the beginning and it creates the expectation for this to work comparably to D^Val_Train.

**Summary Of The Paper:**

This work shows that neural network models trained on datasets affected by the presence of simple spurious correlations also learn “core” features (i.e., features that are truly discriminative and non-spuriously correlated with the training labels). Given this observation the authors propose an algorithm named “Deep Features Reweighing” (DFR) that consists of retaining only the last layer using a “clean” validation set (i.e., a set that is not entirely affected by the spurious correlation present in the training set).

Results are shown on Waterbirds, CelebA, MultiNLI and CivilComments datasets and performance are compared to different states of the art algorithms (Group DRO, JTT, CnC, SUBG, SSA). The results presented show that DFR outperforms many state of the art techniques in terms of worst-group and mean accuracy, and it is on-par with Group DRO. Unlike Group DRO, however, DFR does not require the training set to contain group-information (while this information is required for the validation set for both algorithms).

**Summary Of The Review:**

The work is very interesting and it is worth sharing the findings with the community. While the novelty is limited (see [Kang2019]) the analysis are interesting. The paper needs, however, to either highlight the differences with [Kang2019] or change the focus to be on the application of a known technique (LWS) to this problem (and on the analysis already provided) rather than on presenting a new algorithm. I would be happy to upgrade my rating if the authors could provide an explanation about why DFR is different than LWS (including comparison) or change slightly the focus as just mentioned.

---

> ### Author Response · Authors · 2022-11-17
> **Author Response to Reviewer xHWG (Part 1 / 3)**
>
> Thank you for your detailed and thoughtful feedback! We are glad that you found the paper interesting and the results worth sharing with the community!
>
> The main strengths of our paper are the empirical results on the spurious correlation benchmarks and the analysis: we show that without any modifications to the standard training procedure, neural networks learn a high quality representation of the core features even when they largely rely on spurious features in their predictions. Beyond standard spurious correlation benchmarks, we demonstrate this phenomenon in the extreme simplicity bias scenarios, and in relation to background bias and the texture bias on ImageNet.
>
> We agree that our main contribution is not *algorithmic*, as last layer retraining is quite common in modern deep learning. For example, in Appendix C.4 we say, *“Algorithmically, DFR is a special case of transfer learning…”*.  We propose a simple adaptation of the last layer retraining procedure specifically to the spurious correlation setting, matching or outperforming the existing methods for robustness to spurious correlations. However, there are also several important *algorithmic* differences between DFR and the work of Kang et al., as we describe below.
>
> Following your suggestions, **we updated the paper to explain the connections to other last layer retraining methods, including [Kang2019], clarified our statements in section 4.1, moved the results for $\text{DFR}_{\text{Tr}}^{\text{Tr}}$ to the appendix, and added multiple ablations**.
>
> We now respond to your comments in detail.
>
> ## Differences with LWS from Kang et al. 2019
>
> Before going into detail about the algorithmic differences in the methods, we want to emphasize that our work and Kang et al. consider different problems. Kang et al considers long-tail classification with class imbalance, while we consider spurious correlations and shortcut learning. In our setting, there is often no class imbalance, and the methods of Kang et al. cannot be directly applied. Our conceptual results, such as the ability to control the reliance on background or texture features in trained models are also orthogonal to the observations of Kang et al. Algorithmically, DFR is indeed related to the methods of Kang et al., but there are still important differences.
>
> **LWS.** In the Learning Weight Scaling (LWS) method that you mentioned, the authors rescale the logits of the classifier with scalar weights $f_i$: the weight of the $i$-th row of the weight matrix in the last layer is updated to $\hat{w}_i = f_i w_i$. The parameters $f_i$ are trained by minimizing the loss on the *training set* with *class-balanced sampling*. In particular, LWS is not full last layer retraining, as only one parameter per class is learned.
>
> **cRT.** Kang et al. also proposed a Classifier Re-training (cRT) approach which retrains the last layer  on the *training set* with *class-balanced sampling*. cRT is closer to DFR than LWS, as it retrains all parameters in the last layer.
>
> Now, we discuss **algorithmic differences** of these methods with DFR:
> - In DFR, we use a *group-balanced* subset instead of *class-balanced* sampling. This distinction is important, as in the group robustness setting the classes are often balanced, so LWS and cRT are not applicable to the spurious correlation setting.
> - In DFR, we *subsample* the reweighting dataset to be group balanced instead of using *class-/group-balanced sampling*. Specifically, we produce a dataset where the number of examples in each group is the same, and only use these datapoints. In LWS and cRT, Kang et al., use all the datapoints, but sample them so that each class appears with equal probability. This detail is hugely important, as group-balanced sampling does not produce classifiers robust to spurious correlations (see results below).
> - In DFR, we use *held-out data* for retraining the last layer, and *not the training data*. This is the important distinction between DFR_Tr^Val and DFR_Tr^Tr. As you noted in your review, DFR_Tr^Tr performs poorly compared to DFR_Tr^Val.
> - There are also technical differences in how we train the last layer in DFR: multiple last layer retraining steps and strong $\ell_1$ regularization.
>
> To sum up, both cRT and LWS are not immediately applicable to spurious correlation robustness. Moreover, even if we adapt cRT to the spurious correlation setting, there are still major differences with DFR: (1) subsampling vs group-balanced sampling, (2) held-out data vs training data. We now provide **new ablations** highlighting the importance of our design decisions compared to cRT:

---

> > ### Author Response · Authors · 2022-11-17
> > **Author Response to Reviewer xHWG (Part 2 / 3)**
> >
> > |                                           | Class or group | Reweighting data | Data balancing    | Waterbirds WGA        | CelebA WGA            |
> > | ----------------------------------------- | -------------- | ---------------- | ----------------- | --------------------- | --------------------- |
> > | LWS [Kang et al]                        | Class          | Train            | Balanced Sampling | 40.03 ± 8.07 | 35.55 ± 14.4       |
> > | cRT [Kang et al]                        | Class          | Train            | Balanced Sampling | 74.48 ± 1.5 | 52.88 ±  5.96       |
> > | Ablation                                  | Group          | Train            | Balanced Sampling | 76.48 ± 1.24 | 56.11 ±  4.77       |
> > | Ablation                                  | Group          | Validation       | Balanced Sampling | 89.21 ± 1.04       | 67.66 ± 2.14 |
> > | $\\text{DFR}\_{\\text{Tr}}^{\\text{Tr}}$  | Group          | Train            | Subsampling       | 90.2±0.8              | 80.7±2.4              |
> > | $\\text{DFR}\_{\\text{Tr}}^{\\text{Val}}$ | Group          | Validation       | Subsampling       | **92.9±0.2**              | **88.3±1.1**              |
> >
> > We added a detailed discussion of the differences to the related work and Appendix A, and the empirical comparison to the Appendix G of the paper.
> >
> >
> > ## Other questions.
> >
> > ### 1. Additional ablation studies.
> >
> > **Multiple last layer re-trains.** In DFR, we train logistic regression on group-balanced subsets of validation data multiple times and average the weights of the resulting models. We added ablation on the number of linear models that we average on Waterbirds and CelebA datasets: in particular, we vary the number of logistic regression retrains in {1, 3, 5, 10, 20} and get the following results (we report mean and std of 5 random seeds):
> >
> > |            | 1                     | 3                     | 5                         | 10                        | 20                          |
> > | ---------- | --------------------- | --------------------- | ------------------------- | ------------------------- | --------------------------- |
> > | Waterbirds | 91.21 ± 1.82       | 92.88 ± 0.45       | 91.73 ± 1.25 | 93.13 ± 0.29 | 92.89 ± 0.19 |
> > | CelebA     | 85.09 ±1.49 | 88.64 ± 1.90 | 87.80 ± 1.17 | 88.02 ± 1.82  | 88.37 ± 2.02    |
> >
> > It is beneficial to retrain the last layer more than once, but the improvements diminish as we increase the number of retrains.
> >
> > **Regularization.** We emphasize the importance of $\ell_1$ regularization in spurious correlations benchmarks where the number of last layer features is much higher than the reweighting dataset size (e.g. 2048 vs 500 on Waterbirds); using 20 model retrains and no $\ell_1$ regularization, on CelebA we get WGA $86.03 \pm 0.42$, and on Waterbirds we get WGA $87.72 \pm 0.42.$
> >
> > **We added these results to Appendix C.1**. In Appendix C.1, we also provide other ablations such as retraining multiple layers and the importance of the base model hyper-parameters.
> >
> >
> > ### 2-3. Pre-training on ImageNet.
> >
> > For experiments on Waterbirds and CelebA, we use ResNet-50 models pre-trained on ImageNet, following prior works in the group robustness [Sagawa et al., 2019; Liu et al., 2021; Idrissi et al., 2021], as we explain in Section 4.1 and Section 6. We explore the effects of pre-training on ImageNet and train dataset fine-tuning in Appendix C and Table 6. Waterbirds is a very small dataset with less than 5000 examples in train split, and the model trained from scratch results in poor performance both for the base model and DFR.
> >
> > However, on the CelebA dataset which has more than 160k train examples, the base model trained from scratch achieves strong average performance and worst group accuracy (WGA) of 39% (worse than random guess). After DFR, we get 85% WGA which is only 3% worse than with the base model which was pre-trained on ImageNet. To sum up, on CelebA even the model trained from scratch (without pre-training) learns a high quality representation of the core features, even though it puts more weight on the spurious features.
> >
> > Moreover, for experiments in section 4.2 on Dominoes and ColorMNIST we do not use any pre-training. These experiments also show that models (even trained from scratch) often learn the core features almost as well as models trained on data without spurious correlations. Following your suggestion, **we updated the phrasing of the result in Section 4.1** to address pre-training.

---

> > > ### Author Response · Authors · 2022-11-17
> > > **Author Response to Reviewer xHWG (Part 3 / 3)**
> > >
> > > ### 4. Fine-tuning on target data.
> > >
> > > Fine-tuning the full model on the validation set. Based on your feedback, we add an additional baseline where we finetune the full model (as opposed to just the last layer) on the group-balanced validation set for $10$ epochs with SGD starting from the ResNet-50 checkpoint pre-trained on ImageNet and without training on the corresponding train splits of Waterbirds and CelebA. We achieve $89.3 \pm 1.3\%$ worst group accuracy on Waterbirds and $84.4 \pm 0.5\%$ on CelebA. While these results are good relative to ERM on the standard training set, they are still significantly worse than DFR_Tr^Val (for reference, DFR_Tr^Val with fine-tuning the feature extractor on the training data achieves significantly higher $92.9$% and $88.3$% WGA respectively). **We added these results to Appendix C.1.**
> > >
> > > **DFR without training on the biased train data.** In the Appendix Table 6, we also show the DFR version which uses pre-training on ImageNet but without further fine-tuning on the target train data in the row $\text{DFR}_{\text{IN}}^{\text{Val}}$. With this approach, we achieve 88.7% worst group accuracy on Waterbirds and 73.1% on CelebA. On both datasets fine-tuning the feature extractor on the training data is necessary to achieve the optimal performance.
> > >
> > > ### 5. Size of the reweighting dataset.
> > > For reference, we provide the dataset descriptions including the size for train and reweighting datasets in Appendix Figures 6 and 7. For example, on CelebA we use a group-balanced reweighting dataset of size 700, and on Waterbirds we use around 500 examples for re-training the last layer, while for the ResNet-50 model the dimensionality of last layer representations is 2048. Consequently, the number of features for the last layer is higher than the number of datapoints.
> > >
> > > ### 6. De-emphasizing $\text{DFR}_{\text{Tr}}^{\text{Tr}}$.
> > > Thank you for your suggestion, **we moved the discussion of $\text{DFR}_{\text{Tr}}^{\text{Tr}}$ to the appendix** C.3. As we explained above, the use of held-out data for last layer retraining is one of the important algorithmic differences between DFR and prior work, and leads to significant improvements in performance.
> > >
> > >
> > > ## Summary
> > >
> > > **Based on your feedback, we added several new ablations and explicit comparison to prior work, and updated the presentation.** Given these updates, we hope that you will consider raising your score. Please feel welcome to ask any further questions, and we will respond if the openreview system permits.
> > >
> > >
> > > **References**
> > >
> > > [Kang et al., 2019]
> > > [*Decoupling Representation and Classifier for Long-Tailed Recognition*](https://arxiv.org/abs/1910.09217)
> > >
> > > [Sagawa et al., 2019] [*Distributionally Robust Neural Networks for Group Shifts: On the Importance of Regularization for Worst-Case Generalization*](https://arxiv.org/abs/1911.08731)
> > >
> > > [Liu et al., 2021] [*Just Train Twice: Improving Group Robustness without Training Group Information*](https://arxiv.org/abs/2107.09044)
> > >
> > > [Idrissi et al., 2021] [*Simple data balancing achieves competitive worst-group-accuracy*](https://arxiv.org/abs/2110.14503)

---

> > ### Comment · Reviewer_xHWG · 2022-11-18
> > **One last clarification on cRT (and LWS) and their correct application to spurious correlations**
> >
> > Thanks to the authors for your explanation. I do appreciate that you have addressed many, if not all, my questions and requests and added new ablation studies. And thank you also for correcting me on LWS vs cRT.
> >
> > I would like to understand just one last aspect: can you elaborate more on why you say that cRT (and LWS) are not applicable to spurious correlation? Is it just because in their proposal they used a class-balanced dataset rather than a group-balanced one or are there other limitations?
> >
> > I would say that Kang et al algorithms can be used in both scenarios given the proper choice of balanced dataset. Applying Kang et al proposals using a class-balanced dataset to tackle spurious correlations does not seem the correct comparison. Given that DFR relies on an "oracle" that provides a group-balanced dataset, the same oracle can be used with cRT (LWS).
> >
> > If I am wrong, I'd appreciate your explanation. If I am correct I think the new experiments below with Kang et al should be presented using the same data balanced dataset used in DFR (i.e., the same training and validation sets, given that the selection of the validation dataset is not part of the DFR contribution).
> >
> > To be clear that would be train a model using the same training set, then "correct" the model using cRT (and LWS) using the same validation set used by DFR.

---

> > > ### Author Response · Authors · 2022-11-19
> > > **Thank you for your response & further clarifications**
> > >
> > > Thank you for your response!
> > >
> > > First, we would like to emphasize that **LWS is algorithmically not relevant to the spurious correlations setting**. LWS learns **one parameter per class $f_i$**, and these parameters are used to rescale the class logits. Regardless of the retraining data and data sampling strategy LWS would only address the class imbalance problem. In the  [Response part 2/3](https://openreview.net/forum?id=Zb6c8A-Fghk&noteId=sXyj654LqQl).  we included original LWS from [Kang et al] which used train data class-balanced sampling to learn $f_i$.  Per your request, we also train LWS using **the same validation set** used by DFR and group-balanced sampling and LWS achieves $53.07 \pm 16.44$% WGA on Waterbirds and $51.11 \pm 17.36$% WGA on CelebA which shows that it cannot address the spurious correlations robustness.
> > >
> > > Further, we would like to carefully explain the **differences between cRT and DFR** and emphasize that we provide the results of the cRT modification that uses **the same validation set** for re-training as DFR in the row 4 of the first Table of Response part 2 / 3.
> > >
> > > **Classes vs groups**.
> > > The first difference is that cRT and LWS use class balancing and not group balancing, as they focus on long tail classification and not spurious correlations. We argue that the long tail classification and spurious correlations are qualitatively different tasks. However, there are further algorithmic differences.
> > >
> > > **Train vs held-out data**.
> > > cRT uses train data for re-training the classifier, and this strategy works well for long-tail classification, but not for spurious correlations. We believe that the use of held-out data is an important contribution of our work, as it has not been done in Kang et al, and makes a big difference in performance. In particular, see Appendix C.4 of the submission and [response to reviewer AQfF](https://openreview.net/forum?id=Zb6c8A-Fghk&noteId=gUdthxmK3m) for a detailed explanation of why held-out data is necessary.
> > >
> > > **Balanced sampling vs subsampling**.
> > > cRT uses (class-)balanced sampling, while we use (group-)balanced **subsampling** of the data. The difference between the two approaches is that in balanced sampling each datapoint is sampled with probability inversely proportional to the size of the class (group), while in subsampling we discard the data from the majority groups, and only use the balanced subset of the data. This is an important algorithmic difference that is crucial for strong performance.
> > >
> > > What you call *oracle that provides a group-balanced dataset* is a part of the method which is different between cRT and DFR. More precisely, we can think that the oracle provides the reweighting dataset and the corresponding group labels. As we explained above, cRT and DFR use this information differently (even if we replace classes with groups): cRT uses balanced sampling, while DFR subsamples the dataset by discarding data from majority groups.
> > >
> > > To sum up, in order to convert cRT into DFR, we need to (1) replace classes with groups, (2) replace the training data with held-out data, and (3) replace balanced sampling with subsampling (more precisely, we subsample the dataset multiple times, train linear models with strong $\ell_1$ regularization, and average the weights of these models). **We ablate these 3 design choices in the first table rows 2-4 in**  [Response part 2/3](https://openreview.net/forum?id=Zb6c8A-Fghk&noteId=sXyj654LqQl).
> > >
> > > Each of these design choices is important. In particular, even if we replace classes with groups and the training set with validation in cRT, we still only get 67.7% worst group accuracy on CelebA (row 4 in the Table) instead of 88.3% with DFR, due to balanced sampling instead of subsampling. Both methods use **the same validation dataset** for last layer re-training.
> > >
> > > We believe that all three steps for converting cRT to DFR are non-trivial. While the difference between groups and classes is mostly conceptual, the change of the training data to held out data and the change from balanced sampling to subsampling are **important algorithmic** differences. Moreover, we demonstrate the importance of the number of re-trains and the regularization in [Response part 2/3](https://openreview.net/forum?id=Zb6c8A-Fghk&noteId=sXyj654LqQl).
> > >
> > > Finally, we would like to emphasize that while we make algorithmic contributions, the main strength and contribution of our work is in showing that neural networks learn about core features even in the presence of spurious features, and leveraging this insight specifically in the context of spurious correlations to show that a simple last layer retraining approach outperforms many sophisticated methods for spurious correlations that have been published recently, alongside a thorough empirical evaluation and analysis.
> > >
> > > Please let us know if you have any further questions, and we will be happy to respond if the openreview system allows us to.

---

> > > > ### Comment · Reviewer_xHWG · 2022-11-21
> > > > **Final comment**
> > > >
> > > > Thank you again for your patient explanation. I fully agree with the authors that the main contribution is the analysis and the insight that the paper brings to light. I still believe that the correct comparison however between cRT (and LWS) and DFR is to use the same held out dataset used in DFR. I appreciate that the choice of the dataset and how to split it and use it is part of the algorithm but I do believe it would be a better more candid comparison as it would highlight how much of the importance is due to such choice compared to the small algorithmic differences. That said, this does not change the merit of this submission.
> > > >
> > > > Based on the discussion with the authors I am happy to change my score to accept (I'd still love to see cRT with the same held out set used for DFR, if you can).

---

### Author Response · Authors · 2022-11-17
**Title: General Response to Reviewers and Area Chair**

We thank the reviewers for insightful feedback and support!

We are happy that all reviewers highlighted that our paper is making important timely contributions, advances our understanding of representation learning, and that the results are worth sharing with the community. In particular,
- We show that standard ERM-trained classifiers learn a high quality representation of the core features, even when they primarily rely on spurious features in their predictions.
- We show that this phenomenon holds very broadly, both in vision and in NLP, including extreme simplicity bias scenarios.
- We develop a simple last layer retraining procedure, DFR, which achieves state-of-the-art performance on multiple spurious correlation benchmarks with only one hyper-parameter.
- In addition to the common group robustness benchmarks, DFR can be used to reduce biases and reliance on spurious correlations in large pre-trained models.  For example, we show how we can control the reliance on texture and background in ImageNet classifiers with DFR, in just minutes of training on a single GPU.

We believe that these fundamental insights, combined with a simple and effective approach, provide a very impactful contribution. Moreover, inspired by reviewer comments, we have made multiple updates to the paper and added new experiments, described further below. We hope that the timeliness, significance, and potential for impact, in addition to these new results, can be carefully considered by reviewers in the final assessment.

## Updates to the paper and new experiments

Inspired by reviewer comments, we made multiple updates to the paper and added new experiments. In the paper PDF we highlight the updates in blue.

- *Based on suggestions by reviewers xHWG, 8GtD*: We added an **extended discussion of the differences of our work with Kang et al., and Rosenfeld et al**. We added a new paragraph to the related work, an extended discussion in Appendix A, and an empirical comparison in Appendix G. Our work focuses on a different type of problems (spurious correlations instead of long-tail classification and domain generalization), and there are important algorithmic differences between our methods. We also note that Rosenfeld et al is concurrent work.
- *Based on suggestions by reviewer xHWG*: We added three **new ablations** to Appendix C: (1) importance of retraining the last layer on group-balanced subsets several times, (2) importance of regularization in last layer retraining and (3) comparison to fine-tuning a pre-trained model on the group-balanced validation set directly.
- *Based on suggestions by reviewer xHWG*: We **clarified our statements** in Section 4.1 regarding per-training on ImageNet.
- *Based on suggestions by reviewers xHWG and 8GtD*: We **moved the discussion of  $\text{DFR}_{\text{Tr}}^{\text{Tr}}$ from the main text to Appendix** C.3 to simplify the presentation. We also **removed $\text{DFR}_{\text{Tr}}^{\text{Tr}}$ from Table 2**, and **simplified Table 3** significantly, to make the results easier to read.

---

### Public Comment · ~Jiazhi_Li2 · 2023-08-16
**Role of Reweighting**

Dear authors,

Your work is very interesting. I just have a question when I am trying to understand the logic of the proposed method.

I fully acknowledge that the comprehensive empirical experiments verify core features are learned and upon last layer retraining with a small group-balanced dataset, the impact of spurious correlation is mitigated. My question is that for the features which will be inputted into last layer, how to ensure each dimension is disentangled with another dimension so that by last layer reweighting, the dimensions representing core features will be upweighted. I just think that without explicitly supervision on these features, core features and spurious features are entangled in each dimension, which makes the real causes contributing to the performance improvement remain unknown.

Can you give some insights on this?

Thank you.

---

> ### Author Response · Authors · 2023-08-21
> **Thank you for your question**
>
> Hi Jiazhi,
>
> Thank you for your question and your interest in our paper. Note that in practice we are learning a new last layer for the model, which is not necessarily just upweighting some features. In fact, the last layer retraining does not assume that the core and spurious features are disentangled along the embedding dimensions. The main takeaway is that the spurious and core features can be *linearly separated* with a high accuracy, not that they are disentangled. This conclusion is directly supported by our empirical findings.
>
> The exact reasons *why* the features are linearly separated is an open research question! See our Appendix B.4. where we describe the logit additivity phenomenon.

---

### Decision · Program_Chairs · 2023-01-20

**Decision:**

Accept: notable-top-25%

**Justification For Why Not Higher Score:**

I have another paper in the batch that has better scores and that I would consider more interesting for the general audience but happy to discuss of course.

**Justification For Why Not Lower Score:**

N/A

**Metareview: Summary, Strengths And Weaknesses:**

The authors build on the fact that neural network classifiers can largely rely on simple spurious features to make their predictions and show that, contrary to common belief, even in these cases, the networks still often learn core features associated with the desired attributes of the data. The work shows that a simple last layer retraining can match or outperform state-of-the-art approaches on spurious correlation benchmarks, at a much lower complexity and computational requirements.
The work is novel and well received by all reviewers, it clearly makes a nice addition to ICLR.

**Note From Pc:**

if the above contains the word "oral" or "spotlight" please see: "oral" presentation means -> notable-top-5% and "spotlight" means -> notable-top-25%. As stated in our emails, we are disassociating presentation type from AC recommendations